# Efficient-SAM2: Accelerating SAM2 with Object-Aware Visual Encoding and Memory Retrieval

**Jing Zhang**[1,2], **Zhikai Li**[1,*], **Xuewen Liu**[1,2], **Qingyi Gu**[1,*]

[1]Institute of Automation, Chinese Academy of Sciences
[2]School of Artificial Intelligence, University of Chinese Academy of Sciences
{zhangjing2024, zhikai.li, liuxuewen2023, qingyi.gu}@ia.ac.cn

## Abstract

Segment Anything Model 2 (SAM2) shows excellent performance in video object segmentation tasks; however, the heavy computational burden hinders its application in real-time video processing. Although there have been efforts to improve the efficiency of SAM2, most of them focus on retraining a lightweight backbone, with little exploration into post-training acceleration. In this paper, we observe that SAM2 exhibits sparse perception pattern as biological vision, which provides opportunities for eliminating redundant computation and acceleration: i) In mask decoder, the attention primarily focuses on the foreground objects, whereas the image encoder in the earlier stage exhibits a broad attention span, which results in unnecessary computation to background regions. ii) In memory bank, only a small subset of tokens in each frame contribute significantly to memory attention, and the salient regions exhibit temporal consistency, making full-token computation redundant. With these insights, we propose Efficient-SAM2, which promotes SAM2 to adaptively focus on object regions while eliminating task-irrelevant computations, thereby significantly improving inference efficiency. Specifically, for image encoder, we propose object-aware Sparse Window Routing (SWR), a window-level computation allocation mechanism that leverages the consistency and saliency cues from the previous-frame decoder to route background regions into a lightweight shortcut branch. Moreover, for memory attention, we propose object-aware Sparse Memory Retrieval (SMR), which allows only the salient memory tokens in each frame to participate in computation, with the saliency pattern reused from their first recollection. With negligible additional parameters and minimal training overhead, Efficient-SAM2 delivers $1.68\times$ speedup on SAM2.1-L model with only 1.0% accuracy drop on SA-V test set, where SWR and SMR provide $1.83\times$ and $1.78\times$ speedups, respectively. Code is available at: https://github.com/jingjing0419/Efficient-SAM2.

## 1 Introduction

Segment Anything Model (SAM) (Kirillov et al., 2023) was introduced as a vision foundation model, enabling promptable segmentation following user's instruction (Li et al., 2024b; Chen et al., 2024), and its successor, SAM2 (Ravi et al., 2024; Jiaxing & Hao, 2025), extends this framework with video understanding capabilities. Leveraging its memory mechanism, SAM2 can reference long-term historical information to process streaming video sequences, achieving remarkable performance in video object segmentation (VOS) (Ding et al., 2024) and video object tracking tasks (Cuttano et al., 2025; Yang et al., 2024). However, the reliance on a large-scale visual backbone and frame-level memory interaction incurs prohibitive computational costs and latency (Zhou et al., 2025; Xiong et al., 2024a; Sun et al., 2025), which limits its applicability in real-time video processing.

The primary latency bottlenecks of SAM2 lie in its image encoder and memory attention block. To address this, EdgeTAM (Zhou et al., 2025) distills a lightweight model architecture and introduces

---

*Corresponding authors: {zhikai.li, qingyi.gu}@ia.ac.cn.

a spatial perceiver for memory compression, achieving edge-level efficiency. However, its high training costs and degraded performance hinder practical deployment. A second line of research accelerates vision Transformer (ViT) backbones by dynamically merging similar tokens (Bolya & Hoffman, 2023; Norouzi et al., 2024); yet these approaches not only fail to adapt to SAM2's hierarchical, window-attention-dominated architecture, but also struggle with segmentation tasks (Kienzle et al., 2024), resulting in severe performance degradation and undesirable practical acceleration.

In this work, we observe that SAM2 exhibits a sparse perception pattern as biological vision; however, its default processing fails to capitalize on this sparsity, leading to redundant computation.

❶ **Object-focused attention in mask decoder vs. Broad attention span in image encoder.** During mask decoding, the prompt-to-image attention exhibits object preference towards prompt's interests. As shown in Figure 1, attention is focused on the foreground objects and potential distractions, while the background features are significantly suppressed. In contrast, the image encoder in the earlier stage is unaware of prompt's interest and exhibits board attention span, thus producing useless computations.

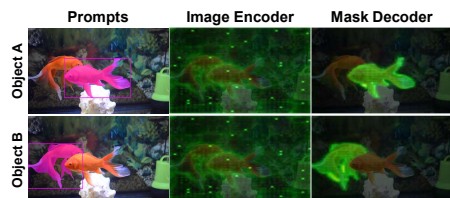

Figure 1: The image encoder exhibits broad attention coverage, but the mask decoder focuses narrowly on prompt-relevant objects.

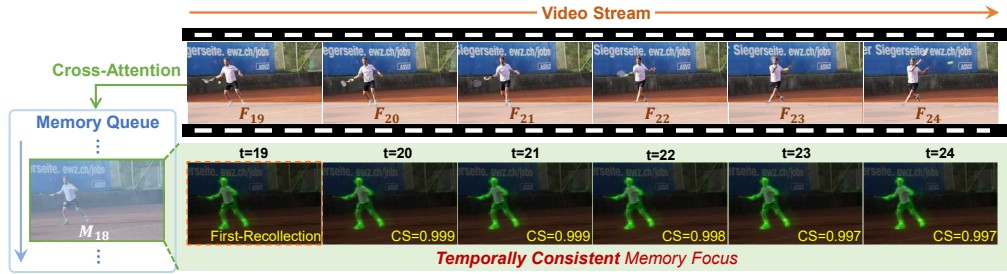

Figure 2: In memory attention, the memory frame exhibits concentrated attention distribution, suggesting redundancy in the memory bank, and its saliency pattern remains temporally consistent, as evidenced by high cosine similarity (CS) to its first recollection.

❷ **Temporally-consistent saliency in memory bank vs. Repeated full-token recollections.** In memory attention, entire memory frames are repeatedly recalled to modulate image features. However, we observe that the useful information is sparsely distributed, with only a small subset of tokens actively providing cues. As shown in Figure 2, in the default mode, a memory frame is recomputed for every incoming frame before it exits the memory queue; yet the attention pattern shows strong temporal consistency, frequently focusing on a fixed set of informative tokens, which provides opportunity to prune unnecessary computation.

With the above insights, we propose Efficient-SAM2, an object-aware computation scheduling and optimization framework to accelerate SAM2, with a particular focus on improving the efficiency of both image encoder and memory attention. Specifically, for image encoder, we introduce object-aware Sparse Window Routing (SWR), which performs computation allocation at the window level and integrates naturally with SAM2's window attention. It assigns background windows to a lightweight shortcut branch to reduce redundant computation, where the router leverages the consistency and saliency cues provided by previous frame's mask decoder. Here, the shortcut branch consists of two linear layers with negligible parameters, which can be trained efficiently. In addition, for memory attention, we propose object-aware Sparse Memory Retrieval (SMR), which selects only the salient tokens in the memory bank to participate in attention computation, effectively reducing the overall token cost. Notably, leveraging temporal consistency, the saliency pattern of a memory frame is cached during its first participation in memory attention and can be reused across subsequent frames without recomputation. Our contributions are summarized as follows:

- We propose Efficient-SAM2, a post-training sparse acceleration scheme for SAM2. It addresses the contradiction between SAM2's sparse perception pattern and its dense computation, highlighting redundant operations in image encoder and memory attention, and thus offering new insights for efficient acceleration.

- With the above insights, we propose dedicated acceleration schemes for both image encoder and memory attention. For image encoder, we allocate computation at the window level by routing background windows to a lightweight shortcut branch. For memory attention, we restrict computation to only the salient memory tokens in each frame, with saliency determined by the memory attention map from their initial participation.

- Efficient-SAM2 shows a superior performance-speed trade-off across multiple VOS benchmarks, while adding negligible additional parameters and incurring only minimal training cost. For instance, Efficient-SAM2 delivers $1.68\times$ end-to-end speedup on SAM2.1-L model with only 1% accuracy drop on SA-V test set; individually, SWR and SMR providing $1.83\times$ and $1.78\times$ speedups, respectively.

## 2 RELATED WORKS

**Segment Anything Model.** SAMs (Kirillov et al., 2023; Carion et al., 2025), owing to their robust segmentation and tracking capabilities, have emerged as versatile foundational models in computer vision. In particular, SAM2 (Ravi et al., 2024) introduces a streaming memory mechanism that infuses long-range historical context into the current frame, further extending its capabilities from image to video. Recently, SAM2 has been widely applied to semi-supervised video object segmentation (Ding et al., 2024), video object tracking (Yang et al., 2024; Videnovic et al., 2025), and referring video object segmentation (Cuttano et al., 2025), consistently outperforming prior task-specific models. However, the large image encoder and computationally intensive memory mechanism in SAM2 impose substantial computational burdens and inference latency (Li & Gu, 2023), severely limiting its widespread deployment in real-world applications (Liu et al., 2024a; Li et al., 2024a).

**Lightweight SAM2.** To reduce the computational cost of SAM2's image encoder and memory mechanism, existing methods have conducted valuable explorations (Lv et al., 2024; Zhang et al., 2025). EfficientTAM (Xiong et al., 2024b) designs a lightweight image encoder and leverages pooling operations within its memory mechanism to reduce computational overhead. EdgeTAM (Zhou et al., 2025) introduces a spatial perceiver module to compress memory and retrains the whole network end-to-end to accommodate this module. Although these methods have achieved significant success, they require costly retraining on large-scale datasets. In contrast, post-training acceleration techniques, which avoid end-to-end retraining, are more flexible and generalizable for Efficient SAM2. Unfortunately, post-training acceleration tailored for SAM2 remains largely unexplored.

**Post-training Acceleration.** Post-training acceleration (Li et al., 2023a; Liu et al., 2025), valued for its low implementation overhead, has been extensively studied across diverse model architectures (Liu et al., 2026; Li et al., 2023b; Liu et al., 2024b). For Vision Transformer (ViT) architectures, ToMe (Bolya et al., 2022) dynamically merges similar tokens using a general matching algorithm, reducing computational redundancy without altering the model structure. ALGM (Norouzi et al., 2024) further introduces a two-stage local-to-global merging strategy for segmentation models, significantly enhancing pixel-level task performance. For Diffusion Transformer (DiT) architectures, ToMe4DM (Bolya & Hoffman, 2023) leverages the temporal characteristics of diffusion, introducing randomness within local windows during token matching to accelerate diffusion models without any retraining. While these methods accelerate ViT and DiT models successfully (Li et al., 2023c; 2022c), they fail on SAM2. The windowed attention in image encoder renders existing token merging and matching strategies incompatible, resulting in suboptimal accuracy and speedup.

## 3 METHOD

### 3.1 PRELIMINARIES

**The Pipeline of SAM2.** SAM2's image encoder $E_{\text{img}}$ adopts Hiera (Ryali et al., 2023; Li et al., 2022a) as the backbone to hierarchically encode multiscale image features, with stages denoted as $\{S_0, S_1, S_2\}$, the image feature $\mathcal{F}_t$ is obtained as follows:

$$\mathcal{F}_t = \{F_t^{s_0}, F_t^{s_1}, F_t^{s_2}\} = E_{\text{img}}(I_t), \tag{1}$$

where $I_t$ is the current frame input. Then $F_t^{s_2}$ serves as image embedding $F_t$, which interact with the memory bank $\mathcal{M}_t$ in the memory attention module $A_{\text{Mem}}$, obtaining memory-conditioned image

feature $F_{M,t}$ as follows:

$$F_{M,t} = \text{MA}(F_t, \mathcal{M}_t). \tag{2}$$

Next, the mask decoder $D$ performs cross-attention between prompt embedding $P_t$ and $F_{M,t}$, with $F_t^{s_0}$ and $F_t^{s_1}$ help to provide high-resolution details, deriving the output $O_t$ as follows:

$$O_t = D(F_{M,t}, F_t^{s_0}, F_t^{s_1}, P_t). \tag{3}$$

Finally, $O_t$ and $F_t^{s_0}$ are fused and encoded as memory in the memory encoder $E_{\text{mem}}$ as follows:

$$M_t = E_{\text{mem}}(F_t^{s_0}, O_t). \tag{4}$$

**Image Encoder with Window Attention.** SAM2 predominantly adopts window attention (Li et al., 2022b; Bolya et al., 2024) in its encoder to capture fine-grained local structure. For a window attention module in layer $l$, denote the inputs as $X^{l-1} \in \mathbb{R}^{1 \times H \times W \times C}$, where $(H, W)$ is the feature height and width, and $C$ is channel dimension, the process is as follows:

$$X_W^{l-1} = \text{WindowPartition}(X^{l-1}; h, w) \in \mathbb{R}^{N_W \times h \times w \times C}, \tag{5}$$

$$X_W^l = \text{MHSA}(X_W^{l-1}) \in \mathbb{R}^{N_W \times h \times w \times C}, \tag{6}$$

$$X^l = \text{WindowUnpartition}(X_W^l; h, w) \in \mathbb{R}^{1 \times H \times W \times C}, \tag{7}$$

where WindowPartition($\cdot$) pads and partitions image features into non-overlapping windows while WindowUnpartition($\cdot$) perform reverse operation. $(h, w)$ denotes the window size, $N_W$ represents the number of windows. MHSA($\cdot$) denotes Multi-Head Self-Attention. Thus, attention is computed locally within the windows, leaving windows mutually independent.

**Mask Decoder.** The mask decoder predicts three candidate masks along with their confidence scores $s_{\text{iou}} \in [0, 1]$ and outputs the mask with the highest score. Additionally, it returns an occlusion score $s_{\text{obj}}$, where $s_{\text{obj}} < 0$ indicates object absence, $s_{\text{obj}} > 0$ indicates presence, and the absolute value $|s_{\text{obj}}|$ depicts the tracking confidence.

**Memory Bank.** At moment $t$, $\mathcal{M}_t$ consists of prompted frames $\mathcal{M}_t^p$ and a First-In-First-Out (FIFO) queue $\mathcal{M}_t^q$ of previous frames (Zhou et al., 2024), represented as:

$$\mathcal{M}_t = \mathcal{M}_t^p \oplus \mathcal{M}_t^q = \mathcal{M}_t^p \oplus \{M_{t-m}, M_{t-m+1}, \ldots, M_{t-1}\}, \tag{8}$$

where $m$ denotes queue length, and $\oplus$ denotes concatenation between sets. In semi-supervised VOS task, the length of $\mathcal{M}_t^p$ keeps 1. Equation 8 is the default setting in which memory sampling interval $\Delta t = 1$. For more general formulation, the queue consists every $\Delta t$ frame plus the last frame, i.e., $\mathcal{M}_t^q = \{M_{t-1}, M_{(n-m-2)\Delta t}, M_{(n-m-1)\Delta t}, \ldots, M_{n\Delta t}\}$. Here $n = \lfloor (t-2)/\Delta t \rfloor$. For simplicity, we present subsequent formulations based on $\Delta t = 1$, while they generalize to other cases.

## 3.2 OBJECT-AWARE SPARSE WINDOW ROUTING

The disparity between the image encoder's broad attention span and the mask decoder's object-focused attention reveals significant encoding redundancy. To address this inefficiency, we propose object-aware Sparse Window Routing, a window-level computation scheduling scheme that routes windows through separate pathways based on their relevance to objects. This divide-and-conquer handling of windows aligns naturally with window attention's locality, thereby effectively reducing redundancy and preserving model performance. Specifically, our method consists of two components: an object-aware router for predicting object-relevant windows, and a lightweight shortcut branch for processing background windows efficiently.

**Object-Aware Router.** Assuming that the video stream is continuous both spatially and temporally, the router accurately predicts object-relevant windows $\mathcal{W}_{\text{obj}}$ following the Spatial-Temporal **Consistency** of Object and Perceptual **Saliency** of Object, formulated as:

$$\mathcal{W}_{\text{obj}} = \mathcal{W}_{\text{pred}} \oplus \mathcal{W}_{\text{salient}}. \tag{9}$$

$\mathcal{W}_{\text{pred}}$ are windows that cover the previous frame's prediction masks, thereby ensuring spatial-temporal continuity of the object, formulated as:

$$\mathcal{W}_{\text{pred}} = \{W_i \mid \exists (x, y) \in W_i, O_{t-1}(x, y) = 1\}, \quad O_{t-1} = \vee_{j=0}^2 O_{t-1,j}. \tag{10}$$

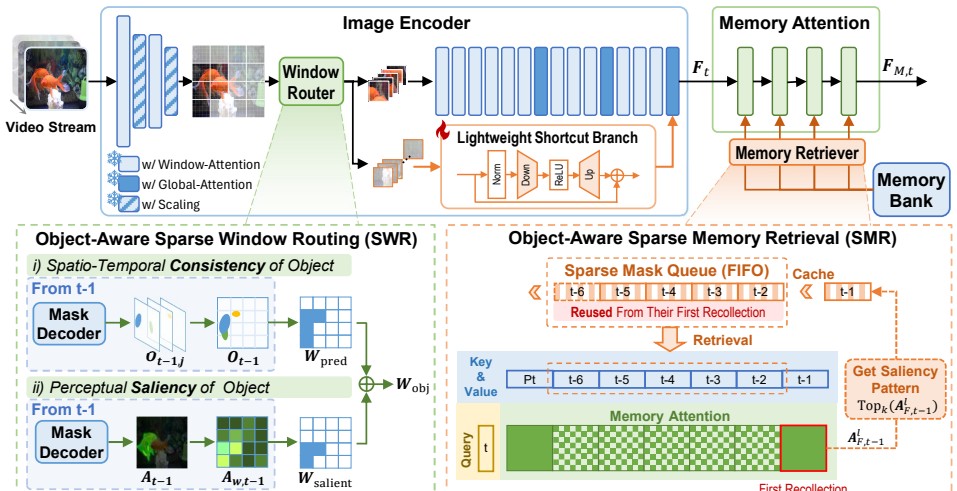

Figure 3: Overview of Efficient-SAM2. For image encoder, we introduce object-aware Sparse Window Routing (SWR), which assigns object-irrelevant background windows to a lightweight shortcut branch based on spatial-temporal consistency and perceptual saliency of the object, thus reducing encoding redundancy. For memory attention, we propose object-aware Sparse Memory Retrieval (SMR), which builds a FIFO mask queue to retrieval most salient memory tokens, in which the saliency patterns are reused from their first recollection, thereby reducing the computational cost.

Here, $O_{t-1,j} \in \{0,1\}^{H \times W}, j = 0,1,2$ are mask predictions. Inspired by SAM2.1++ ([Videnovic et al., 2025](#)), which reveals that the mask decoder could capture distractors in predictions with lower $s_{iou}$, all three alternative masks are utilized to enhance distractor awareness. Moreover, to address motion-induced boundary escape, dilation is applied to $O_{t-1}$ for robust window estimation.

When tracking confidence is low or no mask is predicted, $\mathcal{W}_{salient}$ broadens perception view by adding windows with high cross-attention saliency from previous frame's mask decoder as follows:

$$\mathcal{W}_{salient} = \begin{cases} \{W_k \mid \sum_{\alpha_i \geq \alpha_k} \alpha_i \leq \tau\}, & \text{if } s_{obj} < \theta_{obj}, \\ \emptyset, & \text{otherwise,} \end{cases} \tag{11}$$

where $\theta_{obj}$ is a threshold for prediction confidence, $\tau$ determines the cumulative attention threshold. For each window, its saliency score $\alpha_i \in A_{W,t-1}$ is computed as:

$$A_{W,t-1} = \{\alpha_i \mid \alpha_i = \sum_{(x,y) \in W_i} A_{t-1}[x,y] \in \mathbb{R}^1, i = 1, \dots, N_W\}, \tag{12}$$

where $A_{t-1} \in \mathbb{R}^{H \times W}$ is the average cross-attention weight of image tokens in the mask decoder. This saliency-guided selection protects both re-tracking and anti-interference capabilities, effectively handling challenges including object occlusion, reappearance, and dynamic backgrounds.

**Lightweight Shortcut Branch.** The shortcut branch is designed for efficient background feature processing and alignment. Specifically, it consists of only two linear layers, which is formulated as:

$$X' = X + W_{up}(\text{ReLU}(W_{down}\text{LayerNorm}(X))), \tag{13}$$

where $W_{down} \in \mathbb{R}^{d_r \times d}$ and $W_{up} \in \mathbb{R}^{d \times d_r}$ denote the down-projection and up-projection matrices, respectively, with $d_r = \frac{1}{2}d$. The parameter count of the branch module is merely $d^2 + 2d$, which is substantially smaller than the full transformer block ($12d^2 + 13d$).

The shortcut module is trained through low-cost reconstruction with a small number of training samples. The reconstruction loss function is defined as:

$$\mathcal{L} = \left\| F_M^s - F_M^t \right\|_2^2, \tag{14}$$

where $F_M^t$ is the feature from the original teacher model, and $F_M^s$ is the feature from the student model with the shortcut branch. We adopt memory-conditioned features as the reconstruction target for two key considerations. Directly reconstructing image embeddings would deviate from the

shortcut branch's purpose, which is to provide an efficient path for background feature alignment. Meanwhile, features from the mask decoder are ineffective due to its strong suppression effect on background information. Therefore, we choose memory-conditioned feature for its moderate background information, enabling effective branch learning and robust feature alignment.

### 3.3 OBJECT-AWARE SPARSE MEMORY RETRIEVAL

We empirically observe two properties of memory attention: (i) the sparsity in memory bank: only a small subset of memory tokens receives dominant attention; and (ii) the temporal consistency of saliency pattern: attention distributions over the same memory frame remain highly similar across consecutive video frames queries $\{F_{M,t}, F_{M,t+1}, \ldots, F_{M,t+m}\}$.

Let $A_{t_2,t_1}^l \in \mathbb{R}^{N \times K}$ denote the cross-attention matrix at layer $l$ between image embedding $F_{M,t_2}$ and memory feature $M_{t_1}$, where $N$ is the number of image tokens and $K$ is the number of memory tokens. The temporal consistency property is characterized by:

$$\text{sim}(A_{t,t-1}^l, A_{t+j,t-1}^l) \approx 1, \quad \forall j \in \{1, 2, \ldots, m\}, \tag{15}$$

where $\text{sim}(\cdot, \cdot)$ measures the cosine similarity.

**Saliency Pattern Recognition in First Recollection.** Leveraging these properties, we propose a layer-wise sparse memory retrieval scheme. Upon a memory frame's first recollection in layer $l$, i.e., the cross-attention with $M_{t-1}$ at moment $t$, we compute and cache its sparse pattern once, and subsequently reuse it in later time steps to avoid redundant computation. Specifically, given memory frame $M_{t-1}$ with $K$ tokens, its saliency pattern $S_{t-1}^l$ is identified as follows:

$$S_{t-1}^l = \text{TopK}(A_{t-1}^l, \lfloor (1-s) \times K \rfloor), \tag{16}$$

where $\text{TopK}(v, k)$ returns the indices of the top-$k$ largest elements of vector $v$, $A_{t-1}^l \in \mathbb{R}^K$ represents the averaged attention weights of $M_{t-1}$ in layer $l$ of the memory attention module, and $s$ is the predefined sparsity ratio. Then $S_{t-1}^l$ is added to the cached retrieval queue for memory retrieval in subsequent $(m+1)$ time steps.

**Saliency Pattern Reuse for Memory Retrieval.** Following a FIFO saliency pattern caching scheme, in current frame processing, the retrieval pattern queue $\mathcal{S}_t^l$ is formulated as follows:

$$\mathcal{S}_t^l = \{S_{t-m}^l, S_{t-m+1}^l, \ldots, S_{t-2}^l\}, \tag{17}$$

which is used for sparse memory retrieval and updated at every time steps. Thereby, the sparse memory bank $\mathcal{M}_t^{s_l}$ is obtained as follows:

$$\mathcal{M}_t^{q,s_l} = \{M_{t-m}^{s_l}, M_{t-m+1}^{s_l}, \ldots, M_{t-2}^{s_l}\} \oplus \{M_{t-1}\}, \tag{18}$$

$$M_{t-j}^{s_l} = M_{t-j} \odot \mathbb{I}_{S_{t-j}^l}, j = 2, \ldots, m, \tag{19}$$

where $\mathbb{I}_{S_{t-1}^l} \in \{0,1\}^K$ is the sparse mask constructed from $S_{t-1}^l$, and $\odot$ denotes element-wise multiplication. Consequently, except for the important prompt frame $\mathcal{M}_t^p$ kept unchanged, only the latest memory frame $M_{t-1}$ maintains its complete form for sparsity pattern computation, while all other frames are distilled through layer-adaptive sparse retrieval. Compared to the dense memory bank $\mathcal{M}_t$, the sparse representation $\mathcal{M}_t^{s_l} = \mathcal{M}_t^p \oplus \mathcal{M}_t^{q,s_l}$ significantly reduces the computational complexity from $\mathcal{O}((m+1)NKd)$ to $\mathcal{O}(2NKd + (m-1)Nkd)$ in each memory attention layer, where $k = (1-s)K \ll K$, and $d$ is the feature dimension. Consequently, SMR reduce the memory redundancy as well as preserve important latest memory to ensure performance.

## 4 EXPERIMENTS

### 4.1 EXPERIMENTAL SETTINGS

**Models and Datasets.** Our experiments incorporate SAM2.1-B+ and SAM2.1-L model with memory intervals of 1 and 5, denoted as SAM2.1-B+$_{\triangle t=1}$, SAM2.1-B+$_{\triangle t=5}$, SAM2.1-L$_{\triangle t=1}$, and SAM2.1-L$_{\triangle t=5}$. We follow the standard semi-supervised VOS protocol, where the ground-truth masks on the first frame are available and serve as prompts. For evaluation, we use well-established

Table 1: Performance and speedup comparison of different acceleration methods on the SAM2.1-B+ model with memory interval $\Delta t$=1 and $\Delta t$=5. The evaluation metric is $\mathcal{J}\&\mathcal{F}$. In 'IE' and 'MA' columns, '✓' indicates efficiency optimization for Image Encoder (IE) and Memory Attention (MA) components, while '✗' indicates original setting. Speedup Ratio is reported at component level. '*' denotes the distillation method. Our proposed method achieve significant speedup while maintaining competitive performance with the original SAM2 models.

| Method | IE | MA | Speedup | SAM2.1-B+ $_{\Delta t=1}$ | | | | SAM2.1-B+ $_{\Delta t=5}$ | | | |
| | | | | SA-V test | SA-V val | DAVIS 2017 val | MOSE val | SA-V test | SA-V val | DAVIS 2017 val | MOSE val |
|---|---|---|---|---|---|---|---|---|---|---|---|
| Original | - | - | - | 77.7 | 77.2 | 89.7 | 73.6 | 79.6 | 79.8 | 88.6 | 73.5 |
| ToMe | ✓ | ✗ | 1.36× | 55.3 | 55.0 | 74.5 | 53.5 | 57.5 | 57.2 | 74.3 | 54.0 |
| ALGM | ✓ | ✗ | 1.05× | 71.9 | 70.8 | 88.4 | 67.0 | 74.5 | 73.0 | 87.5 | 66.2 |
| ToMe4DM | ✓ | ✗ | 1.00× | 62.7 | 63.8 | 80.9 | 59.5 | 63.1 | 65.0 | 80.2 | 58.3 |
| SWR(ours) | ✓ | ✗ | **1.69×** | **75.0** | **74.3** | **89.4** | **71.2** | **76.9** | **76.5** | **88.6** | **70.5** |
| MemPool | ✗ | ✓ | **2.14×** | 72.3 | 72.8 | 86.6 | 71.0 | 74.6 | 74.2 | 85.6 | 69.2 |
| SMR-random | ✗ | ✓ | 1.73× | 76.7 | 76.6 | 89.6 | 72.2 | 78.8 | 77.4 | 88.5 | 73.1 |
| SMR-uniform | ✗ | ✓ | 1.78× | 76.8 | 76.6 | 89.6 | 72.3 | 78.6 | 77.4 | 88.5 | 73.1 |
| SMR(ours) | ✗ | ✓ | 1.82× | **77.8** | **77.4** | **89.7** | **73.2** | **79.4** | **78.4** | **88.6** | **73.5** |
| EdgeTAM* | ✓ | ✓ | 1.63× | 72.1 | 71.8 | 86.4 | 69.9 | 73.0 | 73.4 | 86.2 | 69.5 |
| Efficient-SAM2(ours) | ✓ | ✓ | **1.74×** | 75.5 | 74.5 | 89.3 | 70.9 | 76.9 | 76.6 | 88.5 | 70.1 |

Table 2: Results of different acceleration methods on the SAM2.1-L model with memory interval $\Delta t = 1$ and $\Delta t = 5$. Our methods demonstrates effective performance-speed trade-offs.

| Method | IE | MA | Speedup | SAM2.1-L $_{\Delta t=1}$ | | | | SAM2.1-L $_{\Delta t=5}$ | | | |
| | | | | SA-V test | SA-V val | DAVIS 2017 val | MOSE val | SA-V test | SA-V val | DAVIS 2017 val | MOSE val |
|---|---|---|---|---|---|---|---|---|---|---|---|
| Original | - | - | - | 79.8 | 78.3 | 89.9 | 74.5 | 80.9 | 79.7 | 89.9 | 74.7 |
| ToMe | ✓ | ✗ | 1.43× | 47.8 | 72.4 | 67.0 | 44.7 | 48.1 | 49.2 | 67.0 | 42.2 |
| ALGM | ✓ | ✗ | 1.76× | 54.9 | 53.5 | 88.4 | 67.0 | 56.5 | 55.7 | 87.6 | 66.1 |
| ToMe4DM | ✓ | ✗ | 1.14× | 71.7 | 69.6 | 85.0 | 66.1 | 71.7 | 71.6 | 85.6 | 65.2 |
| SWR(ours) | ✓ | ✗ | **1.83×** | **79.0** | **76.4** | **89.9** | **73.4** | **79.8** | **77.7** | **90.7** | **73.4** |
| MemPool | ✗ | ✓ | **2.04×** | 72.6 | 73.2 | 87.8 | 71.3 | 74.4 | 74.9 | 87.2 | 69.8 |
| SMR-random | ✗ | ✓ | 1.75× | 78.9 | 76.6 | 89.8 | 73.8 | 79.7 | 78.2 | 89.9 | 74.4 |
| SMR-uniform | ✗ | ✓ | 1.76× | 78.9 | 77.7 | 89.8 | 73.9 | 79.3 | 78.2 | 89.9 | 74.5 |
| SMR(ours) | ✗ | ✓ | 1.78× | **79.6** | **77.7** | **89.9** | **74.2** | **80.2** | **78.8** | **89.9** | **74.6** |
| Efficient-SAM2(ours) | ✓ | ✓ | **1.80×** | 78.8 | 75.5 | 89.7 | 72.6 | 79.0 | 77.7 | 90.7 | 73.1 |

VOS benchmarks: SA-V (Ravi et al., 2024) test and validation sets, DAVIS 2017 (Pont-Tuset et al., 2018) validation set, and MOSE (Ding et al., 2023) validation set. See Appendix A for details.

**Baselines.** We build three aspects of baselines for comprehensive comparison. First, for visual backbone acceleration, we include token merge methods ToMe (Bolya et al., 2022), ALGM (Norouzi et al., 2024), and ToMe4DM (Bolya & Hoffman, 2023). Second, for efficient memory attention, we implement the memory pooling (Xiong et al., 2024b) method (MemPool), and SMR variants with random and uniform masks (SMR-random and SMR-uniform). Third, we include a distillation-based method EdgeTAM, which compresses both image encoder and memory attention.

**Implementation Details** We adopt the default setting of original SAM2 with SAM2.1 checkpoints and follow standard inference pipeline of VOS. We perform inference on a single NVIDIA A6000 GPU. The details of speed benchmark are provided in Appendix B. For SWR, we set prediction confidence $\theta_{obj} = 5$, and the window saliency threshold $\tau = 0.7$. For shortcut branch learning, we implement a simple reconstruction pipeline based on the inference process to achieve adaptation and alignment with the backbone. This approach requires only 30 unlabeled samples from SA-V train dataset, and completes in approximately 1 hour on an RTX A6000 GPU. For memory retrieval, we apply sparsity ratio $s = 0.95$ to each memory frame across all memory attention layers. With prompt frame and latest frame maintained dense, the overall sparsity ratio achieves $\frac{5s}{7} \approx 0.68$. We provide details of baseline reproduction in Appendix C.

## 4.2 MAIN RESULTS

**Results of SAM2.1-B+ Models.** Table 1 presents performance and speedup comparisons of various acceleration methods on the SAM2.1-B+ model with memory intervals $\Delta t = 1$ and $\Delta t = 5$.

Our methods demonstrate an excellent balance between performance and efficiency. SWR, which focuses on Image Encoder (IE) acceleration, achieves a 1.65× speedup ratio at $\Delta t = 1$ while maintaining strong performance. For example, in SA-V test the performance achieves 75.0, with only 2.7 drop from the original model. In DAVIS, it reaches 89.4, with only 0.3 accuracy drop. SMR, targeting Memory Attention (MA) optimization, delivers a 1.82× speedup while preserving performance comparable to the original model, achieving 77.8 on SA-V test (merely 0.1 drop) and no accuracy drop on DAVIS. Collaboratively, our Efficient-SAM2 provides a 1.71× speedup at $\Delta t = 1$ with minimal performance degradation. In contrast, other methods such as ToMe offer acceleration but suffer significant performance drops, while MemPool achieves higher speedup (2.14×) but with considerable performance loss.

**Results of SAM2.1-L Models.** Table 2 illustrates the performance of acceleration methods on larger SAM2.1-L model. On this larger model, our methods demonstrate even more significant advantages. SWR achieves a 1.80× speedup at $\Delta t$=1, while maintaining performance nearly identical to the original model, with SA-V test score of 79.0 (only 0.8 lower than the original 79.8) and DAVIS score of 89.8 (just 0.1 below the original 89.9). SMR also perform excellently, reaching a 1.78 × speedup with performance metrics close to original model. Our Efficient-SAM2 method provides a 1.75× inference speedup while remaining competitive on key metrics. Notably, at $\Delta t$=5, our SWR method even surpasses than original model's performance on DAVIS datasets (90.7 vs. 89.9), suggesting the potential of suppressing noises.

Across all models and datasets, our experimental results with SMR and its variants (SMR-random and SMR-uniform) provide evidence for token-level sparsity hypothesis in SAM2's memory bank. Ablation studies with various sparsity ratios demonstrate the sparse memory configurations can maintain or even exceed the performance of the original dense memory.

## 4.3 ABLATION STUDY

**Speedup Analysis.** In Figure 4(a), our method accelerates both B+ and L variants without degrading accuracy. For the Large model, we achieve module-wise speedups of up to 1.80× and 1.78×, with an end-to-end speedup of 1.68×. For the Base+ model, the two major modules reach 1.86× and 1.65×, and the overall pipeline gains 1.59×, which shows balanced latency reduction. In end-to-end comparisons (Figure 4(b)), Efficient-SAM2.1-B+ and -L shift the Pareto frontier, delivering markedly lower inference time at comparable or slightly better performance, while alternatives (ToMe, MemPool, EdgeTAM) achieve smaller speedups or incur larger accuracy drops. Thus, our SMR/SWR variants provide a superior speed–performance trade-off, achieving up to 1.8× acceleration without sacrificing segmentation quality.

**Ablation for SWR.** We compare the SWR shortcut variants, as shown in Table 3, which includes straight bypass without additional processing (Identity), attention-based network (Attention), standard Feedforward Network (FFN), and FFN with bottleneck setting (Bottleneck

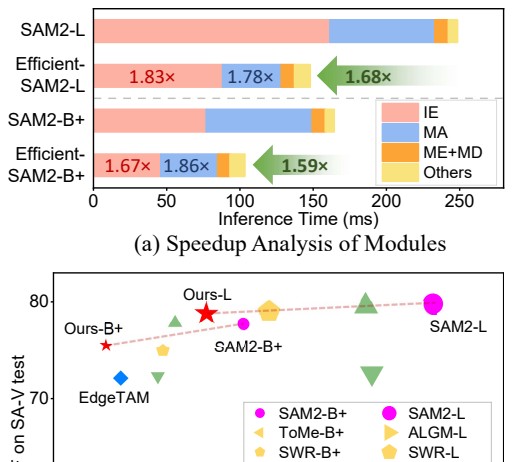

(a) Speedup Analysis of Modules

(b) Speed-Performance Comparison

Figure 4: Detailed speedup analysis. Our method wins a well-balanced accuracy–speed trade-off.

FFN). The three learned shortcut variants yield comparable improvement over Identity at low training cost, underscoring the effectiveness of the proposed training pipeline. Among Bottleneck-FFN retains nearly all achievable accuracy while incurring substantially lower computational and parameter overhead. Moreover, as shown in Table 4, we examine the effects of two enhancement components in $\mathcal{W}_{pred}$ selection, i.e., including all predictions (All Pred.), and performing mask dilation (Dilation). Including all predictions yields modest but consistent gains compared with using

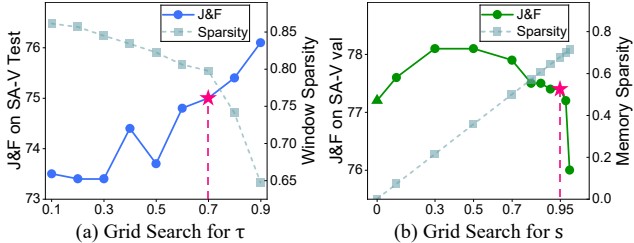

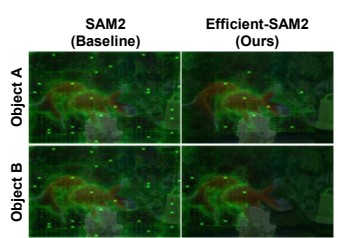

(a) Grid Search for $\tau$     (b) Grid Search for $s$

Figure 5: Sparsity analysis of SWR and SMR. As $\tau$ grows, the window sparsity decreases but stays over 0.6, while the accuracy steadily increases. As the memory sparsity increases, the performance surprisingly surpasses the baseline and then declines,with $s$=0.95 offers a satisfied trade-off.

Figure 6: SWR preserve attention focus on object-relevant regions in the Image Encoder, while blurring attention to unimportant background regions.

only prediction with maximum $s_{\text{iou}}$ (e.g., +0.5 on SA-V test, +0.7 on MOSE), suggesting improved distractor awareness. Dilation produces larger increments (e.g., +1.4 on SA-V test, +1.6 on MOSE).

Table 3: Comparison of shortcut variants in SWR. All trainable branches outperform the Identity, with Bottleneck-FFN offers a parameter-efficient yet effective option.

| Model | SWR Shortcut | Param. (M) | Training Hours | $\mathcal{J}\&\mathcal{F}$ |
|---|---|---|---|---|
| SAM2.1-B+ | Identity | 0 | 0 | 73.0 |
| | Attention | 0.35 | 1.05 | **75.2** |
| | FFN | 0.81 | 1.10 | 75.0 |
| | BottleNeck-FFN | **0.20** | **0.97** | 75.0 |
| SAM2.1-L | Identity | 0 | 0 | 75.7 |
| | Attention | 0.58 | 1.15 | 78.5 |
| | FFN | 1.33 | 1.28 | **79.2** |
| | BottleNeck-FFN | **0.33** | **1.10** | 79.0 |

Table 4: Ablation study of SWR components. Including all predictions (All Pred.) outperforms single prediction; Performing dilation exhibits stable improvement.

| SWR | | SAM2.1-B+ ($\Delta t = 1$) | | | |
|---|---|---|---|---|---|
| All Pred. | Dilation | SA-V test | SA-V val | DAVIS 2017 val | MOSE val |
| ✗ | ✗ | 73.6 | 73.7 | 88.8 | 69.5 |
| ✓ | ✗ | 74.1 | 73.8 | 89.2 | 70.2 |
| ✗ | ✓ | **75.0** | 73.8 | **89.5** | 71.1 |
| ✓ | ✓ | **75.0** | **74.3** | 89.4 | **71.2** |
| Original (Baseline) | | 77.7 | 77.2 | 89.7 | 73.6 |

**Sparsity Analysis.** For SWR, window sparsity is defined as the ratio of the number of $\mathcal{W}_{\text{obj}}$ to total windows (25 for Base+ and 16 for Large), which depends on adaptive object occupancy and the tunable saliency threshold $\tau$. We reports the average window sparsity and the corresponding module speedup across datasets (Table 5), showing the cross-dataset robustness of the module speedup ratio. Furthermore, A grid search for $\tau$ (Figure 5(a)) indicates that

Table 5: Cross-dataset analysis of SWR module. with content-adaptive window sparsity, SWR attains consistent speedups across datasets.

| Dataset | #Videos | SAM2.1-B+ | | SAM2.1-L | |
|---|---|---|---|---|---|
| | | Sparsity | Speedup | Sparsity | Speedup |
| SA-V test | 150 | 0.80 | 3.12× | 0.75 | 3.24× |
| SA-V val | 155 | 0.77 | 2.96× | 0.74 | 3.14× |
| DAVIS 2017 val | 30 | 0.61 | 2.17× | 0.50 | 2.03× |
| MOSE val | 311 | 0.72 | 2.48× | 0.66 | 2.61× |

$\tau$=0.7 provides a balanced trade-off between accuracy and sparsity. For SMR, the memory sparsity is defined as $\frac{5s}{7}$. A grid search for $s$ (Figure 5(b)) shows the accuracy remains above the dense baseline until $s \approx 0.95$, confirming memory redundancy and validating the effectiveness of SWR.

## 5 CONCLUSION

In this paper, we present Efficient-SAM2, a post-training acceleration framework for SAM2 that addresses the mismatch between SAM2's inherently sparse perception patterns and its dense processing pipeline. By examining attention concentration in mask decoder and temporal consistency of salient memory tokens, we identify two key inefficiencies and introduce two lightweight, object-aware components to resolve them. SWR reallocates computation at the window level by routing background windows, identified through previous-frame saliency and decoder-derived consistency signals, into a lightweight shortcut branch, while preserving computation for object-relevant regions. SMR confines memory attention to salient tokens only, caching each memory frame's saliency pattern upon first use and reapplying it across subsequent frames, thereby avoiding redundant recomputation. Experiments across multiple VOS benchmarks demonstrate that Efficient-SAM2 achieves a superior accuracy–efficiency trade-off without expensive retraining or architectural redesign.

ACKNOWLEDGMENTS

This work is supported in part by the Strategic Priority Research Program of Chinese Academy of Sciences under Grant Number XDB1100000; in part by the National Natural Science Foundation of China under Grant Number 62276255; in part by the Postdoctoral Fellowship Program of CPSF under Grant Number GZC20251175.

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

## A    EVALUATION BENCHMARKS

**SA-V** (Ravi et al., 2024) The SA-V dataset consists of 50.9K video clips with 642.6K masklets for video segmentation tasks. Its validation set contains 293 masklets across 155 videos, while the testing set comprises 278 masklets from 150 videos. Both evaluation sets feature challenging scenarios including small objects, occlusions, and reappearing targets, providing a robust benchmark for assessing video segmentation models.

**DAVIS** (Pont-Tuset et al., 2018) is a widely-used benchmark for video object segmentation, consisting of 50 high-quality video sequences for training and 30 sequences for validation. Each sequence is annotated with pixel-accurate ground truth masks, featuring diverse object categories and challenging scenarios such as occlusions and appearance changes. The dataset has evolved through multiple versions, with DAVIS 2017 extending the benchmark to multi-object scenarios.

**MOSE** (Ding et al., 2023) dataset contains 2,149 video clips spanning 36 object categories, with a total of 431,725 high-quality segmentation masks. Specifically designed for complex motion scenarios, it includes 5,200 annotated objects. The dataset has demonstrated its value through extensive evaluation of state-of-the-art methods.

## B    SPEEDUP BENCHMARK

The speedup evaluation are conducted on an RTX A6000 GPU at Float32 precision. For fair comparison, we use 20 videos randomly sampled from SA-V test set, including single object and multi-object cases. We choose float32 as the default precision for speed evaluation rather than bfloat16 (BF16) because, on our tested hardware and software stack, the actual acceleration of the BF16 path falls far short of theoretical expectations, yielding little to no end-to-end gains even under aggressive sparsification. For example, even after dropping about 80% of windows and 68% of memory tokens, the end-to-end inference time remains essentially unchanged. We attribute this to the pipeline being dominated by GPU-side irregular memory access and control-flow overheads, rather than dense operators that can be efficiently accelerated by tensor cores. To ensure a fair comparison and to highlight the genuine end-to-end benefits of algorithmic sparsification, we therefore report speed with float32 as the baseline.

We provide module-level speedup for all baselines in Table 1 and Table 2, where the speedup are computed based on efficiency-optimized module. For example, the reported IE speedup is the Image encoder inference time relative to original model, including the additional computational cost introduced by methods. For some methods, e.g., ALGM, ToMe4DM, the speedup results fall far below theoretical expectations. We hypothesize that this situation arises because the additional computation introduced increases the CPU scheduling overhead, resulting in insignificant acceleration for GPU computation bottlenecks. Specifically, we observed that EdgeTAM's VOS inference pipeline differs from the previous testing methodology. To ensure a fair comparison, we separately evaluated the acceleration ratio relative to the original model at BFloat16 precision. The Table 1 reports module-level acceleration ratios, specifically the inference time ratios of the image encoder, memory attention, and the introduced memory perceiver compared to the original SAM2-Base+ model.

We also shows end-to-end speed analysis in Figure 4, where the number of EdgeTAM is converted based on the speedup ratio to align with the numerical ranges of other methods. We attribute this gap not to insufficient GPU compute, but to pipeline overheads introduced by routing/merging and sparsification: the workload shifts from compute-bound dense kernels to memory- and control-bound execution with irregular indexing and scatter–gather.

Table 5 shows fine-grained sub-module level speedup across different dataset. As shown in Figure 3, our SWR accelerate only deeper final stage layers of image encoder, which account for 60% of total time overhead in the image encoder. This metric provides the most direct reflection of how sparsity affects the acceleration ratio.

## C    Implementation Details of Baselines

In our experiments, we implemented and evaluated various acceleration methods on SAM2.1 models with different configurations. For all experiments, we maintained consistent settings across different memory intervals ($\Delta t$=1 and $\Delta t$=5) to ensure fair comparisons.

- **ToMe**: We implemented ToMe with different pruning ratios, 0.2 for SAM2.1-B+ and 0.1 for SAM2.1-L for comparable speedup ratio. To adapt to segmentation task, we prunes a fixed proportion of tokens at each layer and recover spatial structure through unmerge operations in the final layer.

- **ToMe4DM**: We set the pruning ratio to 0.5 for this method. However, its acceleration effect was limited because ToMe4DM requires matching, merge, and unmerge operations at each layer, introducing significant computational overhead. This explains why it only achieved a $1.00\times$ speedup on SAM2.1-B+, effectively providing no acceleration.

- **ALGM**: We implemented ALGM following its original two-stage merging approach for segmentation models, which includes local conditional pooling and global merging with adaptive pruning rates, followed by unmerging to restore spatial structure. We set the similarity threshold for local merging to 0.7. Unlike the original method, we performed two rounds of global merging to increase the pruning rate. Despite these optimizations, ALGM showed limited acceleration in our experiments (only $1.05\times$ on SAM2.1-B+).

- **MemPool**: For the memory pooling method, we configured the pooling kernel size as $2\times2$. While this method achieved the highest speedup ratios ($2.14\times$ on SAM2.1-B+ and $2.04\times$ on SAM2.1-L), it also showed notable performance degradation, particularly on DAVIS and SA-V test sets.

## D    Qualitative Comparison and Analysis

We show a tracking instance with different prompt in Figure 7. Efficcient-SAM2 show comparable performance with original SAM2 model. Figure 7(c) shows the contribution of dilation operation, where motion induced boundary escape leads to excessively focused attention on ambiguous area, degrading attention to other important area.

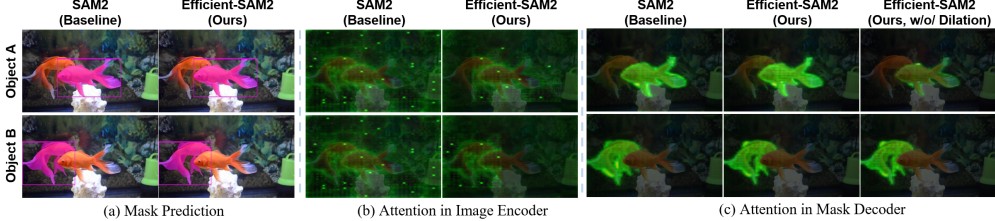

Figure 7: Qualitative comparison and analysis. (a) shows Efficient-SAM2's comparable segmentation quality with original SAM2 model. (b) and (c) illustrate the object-focusing effect of proposed SWR. while SWR blurs background regions, it preserves object-relevant responses in the image encoder, and maintains consistent attention focus in the mask decoder.

## E    Further Discussion of Efficient-SAM2

In our SWR, window routing is used to accelerate the deeper backbone (i.e., stage 2), achieving an excellent accuracy–efficiency balance. We also attempted to extend the SWR scheme to accelerate the shallow stages of inference (which account for about 40% of the backbone's compute latency) and found that SWR can also adapt to the window structure in shallow layers. As shown in Table 6, accelerating more layers yields greater efficiency gains; however, due to the lack of initial aggregation for background windows, this variant incurs a more pronounced accuracy drop, so we did not adopt it. Nevertheless, we believe that with an improved design of the bypass network, this variant has the potential to further recover accuracy, which we leave for future work.

Table 6: Results of Efficient-SAM2 variants. † denotes extend SWR to all encoder stages, which provides more efficiency gain but compromises performance.

| Method | IE | MA | SAM2.1-B+$_{\Delta t=1}$ | | | | | SAM2.1-L$_{\Delta t=1}$ | | | | |
| --- | --- | --- | --- | --- | --- | --- | --- | --- | --- | --- | --- | --- |
| | | | SA-V test | SA-V val | DAVIS 2017 val | MOSE val | Speedup | SA-V test | SA-V val | DAVIS 2017 val | MOSE val | Speedup |
| Original | - | - | 77.7 | 77.2 | 89.7 | 73.6 | - | 79.8 | 78.3 | 89.9 | 74.5 | - |
| SWR | ✓ | ✗ | **75.0** | **74.3** | **89.4** | **71.2** | 1.69 | **79.0** | **76.4** | **89.9** | **73.4** | 1.83 |
| SWR† | ✓ | ✗ | 74.0 | 72.0 | 88.3 | 69.3 | **2.62** | 77.4 | 75.0 | 89.7 | 73.2 | **3.17** |
| Efficient-SAM2 | ✓ | ✓ | **75.5** | **74.5** | **89.3** | **70.9** | 1.74 | **78.8** | **75.5** | **89.7** | 72.6 | 1.80 |
| Efficient-SAM2† | ✓ | ✓ | 74.2 | 72.8 | 88.0 | 69.5 | **2.39** | 76.5 | 74.9 | 89.7 | **72.8** | **2.67** |

## F SHORTCUT TRAINING DETAILS.

We present detailed training settings of shortcut branch in Table 7.

Table 7: Training details of shortcut branch.

| config | value |
| --- | --- |
| Data | Selected from SA-V train |
| #Videos | 30 |
| Video Length | <300 |
| Optimizer | AdamW |
| Sampling Stride | 3 for base+, 4 for Large |
| Learning Rate | 0.0001 |
| Total Steps | 495 |
| Epochs | 3 |
| Update stride | 32 |
| Memory stride | $\{1,3,5\}$ for epoch 0,1,2 |

