# OpenReview forum: "Efficient-SAM2: Accelerating SAM2 with Object-Aware Visual Encoding and Memory Retrieval"
_ICLR.cc/2026/Conference — ICLR 2026 Poster_

### Official Review · Reviewer_FsUz · 2025-10-31

**Soundness:** 3
**Presentation:** 2
**Contribution:** 2
**Rating:** 4
**Confidence:** 3

**Summary:**

The paper addresses the key motivation of resolving SAM2’s high computational burden that limits its real-time application in video object segmentation (VOS). The  proposed Efficient-SAM2, a post-training acceleration framework, has two core components: 1) Object-Aware Sparse Window Routing (SWR) for the image encoder, which leverages spatial-temporal consistency and perceptual saliency from the previous frame’s mask decoder to route background windows to a lightweight shortcut branch  and preserve full computation for object-relevant windows,2) Object-Aware Sparse Memory Retrieval (SMR) for memory attention, which caches each memory frame’s saliency pattern during its first recollection and reuses it in subsequent frames to only involve salient tokens in computation. Experimentally, Efficient-SAM2 achieves a 1.68× end-to-end speedup on the SAM2.1-L model with only a 1.0% accuracy drop on the SA-V test set.

**Strengths:**

1.  Efficient-SAM2 avoids expensive full-model retraining. It adds negligible parameters  and low training overhead, making it flexible for low computational deployment.
2.  By aligning with SAM2’s natural sparse perception, it eliminates redundancy without compromising core functionality—unlike generic token-merging methods (e.g., ToMe) that cause severe accuracy drops.
3.  SWR (image encoder) and SMR (memory attention) are independent modules, allowing separate optimization or integration with other SAM2 variants. Their design (window-level routing, cached saliency patterns) is intuitive and supported by qualitative analysis (e.g., Figure 6 shows SWR preserves object attention).
4.  It maintains strong accuracy across diverse VOS datasets (SA-V, DAVIS 2017, MOSE) and scales to larger models (SAM2.1-L).

**Weaknesses:**

1. The claimed contribution 1 should be merge with contribution 2 as whole one.


2. SWR relies on hyperparameters like the prediction confidence threshold (θₒᵦⱼ=0.5) and saliency threshold (τ=0.7), while SMR depends on the sparsity ratio (s=0.95). The paper does not explore how these parameters generalize to edge cases (e.g., highly cluttered scenes, fast-moving objects) or different datasets.


3. The variable symbols cause confusions, especially for different A.


4. The codes released as supplementary material fail to run according to SAM-2 environment. The clarity of the code is also limited.

**Questions:**

1.What's the actual theory that the equation (11) can reflect the saliency of a window? It should be explained.

 2.How can the object-aware router adapt to the fast moving object since all the information of prediction and saliency are from the preceding frame?

 3.Mask decoder module seems to disappear from the figure 3. How is the segmentation mask produced?

 4.What is the alignment of  background feature processed by lightweight shortcut branch?

 5.How would Efficient-SAM2 adapt to dynamic prompts (e.g., user-added points in middle frames) that change the focus of attention?

 6.Can you provide statistical data (e.g., average CS across all frames in SA-V/DAVIS) to quantify how often saliency patterns remain consistent? What is the impact of inconsistent patterns on SMR’s accuracy?

---

> ### Author Response · Authors · 2025-11-19
>
> Thank you for the detailed comments. We have carefully considered all suggestions and provide clarifications below.
>
> ---
>
> >W1: Merge contributions.
> >
> Thanks for your suggestion. Our first contribution is our experimental findings on the internal sparsity of SAM2, revealing opportunities for reducing computational redundancy. Motivated by these findings, our second contribution is proposing efficiency optimization methods both image encoder and memory attention modules. We appreciate your feedback and will merge these two contributions, with an improved and more polished description in the revised manuscript.
>
> >W2: Robustness of hyperparameter.
> >
> Thank you for valuable feedback.
>
> **Clarifying the role of these hyperparameters**. The sparsity factor $s$ controls memory reduction, while $\tau$ and $\theta_{obj}$ are designed to handle edge cases by including highly-attended windows. Notably, the confidence score $s_{obj}$ naturally indicates tracking difficulty, with higher values signaling challenging scenarios like cluttered scenes or fast motion.
>
> **We performed grid searches for $\tau$ and $s$ in Figure 5**, demonstrating robustness across a wide range of parameter values as well as the accuracy-efficiency trade-off. To further address your concern, we have added: (1) evaluation results on additional datasets to validate cross-dataset generalization to , and (2) a grid search analysis for $\theta_{obj}$ to demonstrate its robustness in edge cases.
>
> | $\tau$ | 0.1 | 0.2 | 0.3 | 0.4 | 0.5 | 0.6 | **0.7** | 0.8 | 0.9 | Baseline |
> |:----:|:----:|:---:|:---:|:---:|:---:|:---:|:---:|:---:|:---:|:----:|
> | **$J\\&F$ on MOSE** | 71.1 | 71.1 | 71.1 | 71.3 | 70.9 | 71.2 | 71.2 | 71.5 | 72.3 | 73.6 |
> | Window Sparsity | 0.77 | 0.76 | 0.76 | 0.75 | 0.74 | 0.73 | 0.72 | 0.68 | 0.61 | 0.00 |
> | **$J\\&F$ on SeCVOS** | 55.2 | 57.1 | 56.8 | 56.8 | 57.0 | 57.9 | 57.5 | 56.9 | 58.4 | 57.4 |
> | Window Sparsity | 0.76 | 0.74 | 0.72 | 0.71 | 0.69 | 0.66 | 0.63 | 0.58 | 0.48 | 0.00 |
>
>
> | $s$ | 0.1 | 0.3 | 0.5 | 0.7 | 0.8 | 0.85 | 0.9 | **0.95** | 0.98 | Baseline |
> |:-:|:----:|:----:|:----:|:----:|:----:|:----:|:----:|:----:|:-----:|:----:|
> | **$J\\&F$ on MOSE** | 73.65 | 73.68 | 73.65 | 73.5 | 73.48 | 73.45 | 73.4 | 73.2 | 72.99 | 73.6 |
> | Token Sparsity | 0.07 | 0.21 | 0.36 | 0.50 | 0.57 | 0.61 | 0.64 | 0.68 | 0.70 | 0 |
> | **$J\\&F$ on SeCVOS** | 57.6 | 57.5 | 57.8 | 57.7 | 57.4 | 58.0 | 58.1 | 59.0 | 53.7 | 57.4 |
> | Token Sparsity | 0.07 | 0.21 | 0.36 | 0.50 | 0.57 | 0.61 | 0.64 | 0.68 | 0.70 | 0 |
>
>
> | $\theta_{obj}$          | 1      | 3      | 5      | 7      | 9      |
> |:-----:|:------:|:------:|:------:|:------:|:------:|
> | **$J\\&F$ on SAV_test**     | 74.7   | 75.1   | 75     | 75.2   | 75     |
> | Window Sparsity         | 0.781  | 0.781  | 0.781  | 0.781  | 0.781  |
> | **$J\\&F$ on SeCVOS**       | 57.8   | 57.6   | 57.5   | 58     | 58.1   |
> | Window Sparsity         | 0.632  | 0.632  | 0.630  | 0.631  | 0.629  |
> | **$J\\&F$\* on MOSEv2**      | 42.11  | 42.26  | 42.2   | 42.82  | 42.4   |
> | Window Sparsity         | 0.796  | 0.795  | 0.795  | 0.797  | 0.797  |
>
> >W3: Variable symbols cause confusions.
> >
> Thank you for your suggestion. We will provide clearer explanations for symbols. Specifically, for the symbol A:
>
> - In Eq. (2), $A_{M}$ represents memory attention module. We will revise the notation from $A_{M}$ to $\text{MA}$ for better clarity.
>
> - In Eq. (12), $A_{t-1}$ is the average cross-attention weight of image tokens across mask decoder layers. $A_{W, t-1}$ represents the sum of $A_{t-1}$ within each window.
>
> - In Eq. (15), $A_{t, i}^{l}$ denote the cross-attention matrix at layer $l$ between image embedding $F_{M,t}$ and memory feature $M_{t−i}$.
>
> - In Eq. (16), $A_{t-1}^{l}$ represents the averaged attention weights of $M_{t−1}$ in $l$-th memory attention layer
>
> >W4: Code issues.
> >
> Thanks for your feedback. We have verified that the code runs successfully in the SAM-2 environment on three different machines. For your reference, we provide environment configurations:
>
> ```
> antlr4-python3-runtime    4.9.3
> gitpython                 3.1.44
> ipython                   8.37.0
> nvidia-cuda-cupti-cu12    12.1.105
> nvidia-cuda-nvrtc-cu12    12.1.105
> nvidia-cuda-runtime-cu12  12.1.105
> opencv-python             4.11.0.86
> python                    3.10.18
> python-dateutil           2.9.0.post0
> python-json-logger        3.3.0
> sam-2                     1.0
> torch                     2.5.1+cu121
> torchaudio                2.5.1+cu121
> torchvision               0.20.1+cu121
> types-python-dateutil     2.9.0.20250516
> ```
> We suspect there might be some differences in the environment setup. If you could provide specific error messages or details about the issues encountered, we would like to assist you in resolving them.
> To improve code clarity, we will add detailed comments and a comprehensive README with step-by-step instructions.

---

> ### Author Response · Authors · 2025-11-19
>
> >Q1: Explanation of equation (11).
> >
> Thanks for your feedback. Equation (11) describes our strategy for selecting windows based on their attention-derived saliency when the confidence score falls below a threshold. Specifically, in Eq. (11), $\alpha_i$ denotes the sum of attention scores for window $i$, which serves as the saliency score of that window. $\sum\limits_{\alpha_i \geq \alpha_k} \alpha_i \leq \tau$  selects the top-$k$ windows whose cumulative saliency scores reach the threshold $\tau$.
> We use cross-attention scores from the mask decoder as a saliency proxy, which reflect how much the prompt attends to different image regions (as shown in Figure 1).
> To accommodate varying attention distribution, we use a cumulative scoring approach rather than a fixed number of windows, enabling adaptive selection.
>
> >Q2: Adaptability of SWR to fast moving objects.
> >
> Thank you for bringing up this important question.
> While the window router does rely on the previous frame's prediction and saliency, it is important to note that windows assigned as background are not discarded. Instead, they are processed through a shortcut network and aggregated with foreground windows in the final layer, thus allowing the model to maintain a global view throughout the processing pipeline.
>
> Furthermore, SWR specifically designed two strategies to address challenging scenarios such as fast motion and severe distraction: (1) **mask dilation**, which helps capture motion-escaping objects that might move beyond their predicted regions; (2) **involving all three predictions** instead of only the mask with the highest IoU score, which leverages SAM2's inherent ability to perceive and handle distractions. **Ablation studies in Table 4** validate the effectiveness of these strategies.
>
> Moreover, we further evaluate SWR's adaptability to more challenging benchmarks including MOSEv2[1] and SeCVOS[2], which contain more instances of fast motion, as well as scenarios involving occlusion, disappearance-reappearance, scale variation, and crowding.
>
> **Results on MOSEv2 Benchmark:**
>
> | Model | Method | IE | MA | $J\\&\dot{F}$ | $J\\&\dot{F}_d$ | $J\\&\dot{F}_r$ | $J\\&F$ |
> |:----:|:----:|:----:|:----:|:-----:|:----:|:----:|:----:|
> | SAM2.1-B | Ori | - | - | 43.29 | 60.59 | 20.73 | 44.38 |
> | | ToMe | $\checkmark$ | $\times$ | 20.07 | 48.49 | 2.00 | 20.82 |
> | | **SWR (ours)** | $\checkmark$ | $\times$ | 42.82 | 60.40 | 20.13 | **43.9** |
> | | MemPool | $\times$ | $\checkmark$ | 40.86 | 61.23 | 18.32 | 41.72 |
> | | **SMR (ours)** | $\times$ | $\checkmark$ | 43.44 | 58.04 | 21.94 | **44.56** |
> | | **Efficient-SAM2 (ours)** | $\checkmark$ | $\checkmark$ | 42.23 | 58.10 | 19.54 | **43.36** |
> | SAM2.1-L | Ori | - | - | 49.56 | 63.34 | 27.90 | 50.81 |
> | | ToMe | $\checkmark$ | $\times$ | 20.38 | 61.21 | 1.62 | 20.97 |
> | | **SWR (ours)** | $\checkmark$ | $\times$ | 45.44 | 57.73 | 23.13 | **46.67** |
> | | MemPool | $\times$ | $\checkmark$ | 43.29 | 62.44 | 21.22 | 44.19 |
> | | **SMR (ours)** | $\times$ | $\checkmark$ | 48.70 | 62.19 | 27.27 | **49.98** |
> | | **Efficient-SAM2 (ours)** | $\checkmark$ | $\checkmark$ | 44.96 | 57.05 | 22.04 | **46.20** |
>
> **Results on SeCVOS Benchmark:**
>
> | Model | Method | IE | MA | No SC | Single SC | Multi SC | **Overall** |
> |:-----:|:----:|:----:|:----:|:----:|:----:|:----:|:----:|
> | SAM2.1-B | Ori | - | - | 77.4 | 58.5 | 51.8 | 57.4 |
> | | ToMe | $\checkmark$ | $\times$ | 48.9 | 15.4 | 12.4 | 18.2 |
> | | **SWR (ours)** | $\checkmark$ | $\times$ | 76.9 | 59.3 | 53.4 | **57.5** |
> | | MemPool | $\times$ | $\checkmark$ | 73.3 | 54.8 | 52.3 | 56 |
> | | **SMR (ours)** | $\times$ | $\checkmark$ | 78.4 | 61.1 | 53.3 | **59.0** |
> | | **Efficient-SAM2 (ours)** | $\checkmark$ | $\checkmark$ | 77.7 | 59.3 | 53.7 | **58.3** |
> | SAM2.1-L | Ori | - | - | 79.3 | 57.9 | 52.4 | 58.0 |
> | | ToMe | $\checkmark$ | $\times$ | 49.1 | 17.7 | 15.2 | 20.4 |
> | | **SWR (ours)** | $\checkmark$ | $\times$ | 78.1 | 55.1 | 53.2 | **56.9** |
> | | MemPool | $\times$ | $\checkmark$ | 74.8 | 55.9 | 51.2 | 55.7 |
> | | **SMR (ours)** | $\times$ | $\checkmark$ | 79.5 | 59.1 | 53.0 | **58.7** |
> | | **Efficient-SAM2 (ours)** | $\checkmark$ | $\checkmark$ | 78.4 | 57.8 | 51.3 | **56.9** |
>
> >Q3: Process of producing segmentation mask.
> >
> Thanks. The segmentation mask is produced by the mask decoder. Specifically, the memory-conditioned image feature ($F_{M,t}$ in Figure 3) interacts with the prompt embedding through cross-attention. This is then fused with the multi-scale features (${F_{t}^{s_0}, F_{t}^{s_1}}$) from the image encoder to generate the final segmentation result $O_t$. We will further improve Figure 3 to make this process clearer.
>
> [1] MOSEv2: A More Challenging Dataset for Video Object Segmentation in Complex Scenes
>
> [2] SeC: Advancing Complex Video Object Segmentation via Progressive Concept Construction

---

> > ### Author Response · Authors · 2025-11-19
> >
> > >Q4: Clarify the feature alignment of shortcut branch.
> > >
> > Thanks. In our SWR, the background windows skips the intermediate layers, which may result in a representation gap between the background features and the deeply processed foreground features. Nevertheless, these features are concatenated for subsequent interactions (global attention layer, mask decoder, memory attention). To address this misalignment, we train a lightweight shortcut branch for background processing.
> >
> > **As shown in the ablation study (Table 3)**, compared to not performing alignment (i.e., using identity mapping), the lightweight shortcut branch is more effective in maintaining model performance.
> >
> > > Q5: Adaption to dynamic prompts.
> > >
> > Thanks. For cases where users add prompts in middle frames, the window router treats that frame as a restart frame and assigns all windows as foreground. From the next frame onwards, window assignment proceeds as usual based on the mask and perception from the previous frame, enabling stable tracking. We will provide a demo and qualitative results for dynamic prompts when the code is released.
> >
> > > Q6: Statistical analysis of saliency patterns consistency.
> > >
> > Thank you for this valuable suggestion. We have quantified the cross-temporal consistency of saliency patterns through statistical analysis across multiple datasets, as summarized in the table below. The **mean CS** metric measures the mean cosine similarity between attention scores when a memory frame first participates in computation versus subsequent computations.
> >
> > The statistical data show that the saliency patterns exhibit high temporal consistency, with the mean CS averaging above 0.9 across all datasets. Moreover, the distribution is relatively concentrated, with the average standard deviation not exceeding 0.08.
> >
> > | Model | Layer | SA-V test |        |        |         | MOSE   |        |        |         | SeCVOS |        |        |         | DAVIS  |        |        |         | **Avg. mean** | **Avg. std** |
> > |:-----:|:-----:|:---------:|:------:|:------:|:-------:|:------:|:------:|:------:|:-------:|:------:|:------:|:------:|:-------:|:------:|:------:|:------:|:-------:|:-------------:|:------------:|
> > |       |       | mean      | std    | min    | 5% pct. | mean   | std    | min    | 5% pct. | mean   | std    | min    | 5% pct. | mean   | std    | min    | 5% pct. |               |              |
> > | SAM2.1-B+ | 0     | 0.9896    | 0.0089 | 0.8802 | 0.9764  | 0.9818 | 0.0177 | 0.8908 | 0.9418  | 0.9654 | 0.0436 | 0.6580 | 0.8700  | 0.9912 | 0.0151 | 0.7663 | 0.9721  | **0.9820**    | **0.0213**   |
> > |       | 1     | 0.9589    | 0.0392 | 0.5836 | 0.8976  | 0.8990 | 0.0772 | 0.6185 | 0.7410  | 0.8791 | 0.1257 | 0.3749 | 0.6052  | 0.9344 | 0.0553 | 0.4045 | 0.8452  | **0.9179**    | **0.0744**   |
> > |       | 2     | 0.9990    | 0.0021 | 0.9479 | 0.9970  | 0.9968 | 0.0047 | 0.9597 | 0.9882  | 0.9951 | 0.0079 | 0.9238 | 0.9788  | 0.9988 | 0.0021 | 0.9790 | 0.9959  | **0.9974**    | **0.0042**   |
> > |       | 3     | 0.9967    | 0.0103 | 0.7513 | 0.9900  | 0.9882 | 0.0176 | 0.8732 | 0.9554  | 0.9691 | 0.0539 | 0.3047 | 0.8663  | 0.9949 | 0.0137 | 0.7212 | 0.9780  | **0.9872**    | **0.0239**   |
> > | SAM2.1-L | 0     | 0.9998    | 0.0005 | 0.9882 | 0.9992  | 0.9994 | 0.0010 | 0.9950 | 0.9973  | 0.9980 | 0.0044 | 0.9074 | 0.9901  | 0.9997 | 0.0012 | 0.9763 | 0.9987  | **0.9992**    | **0.0018**   |
> > |       | 1     | 0.9848    | 0.0195 | 0.7262 | 0.9554  | 0.9588 | 0.0378 | 0.8010 | 0.8789  | 0.9437 | 0.0680 | 0.5650 | 0.7911  | 0.9605 | 0.0360 | 0.6057 | 0.8926  | **0.9620**    | **0.0403**   |
> > |       | 2     | 0.9639    | 0.0333 | 0.6748 | 0.9052  | 0.9301 | 0.0577 | 0.6309 | 0.8148  | 0.9005 | 0.1052 | 0.4214 | 0.6569  | 0.9342 | 0.0550 | 0.5086 | 0.8315  | **0.9322**    | **0.0628**   |
> > |       | 3     | 0.9544    | 0.0394 | 0.4821 | 0.8828  | 0.9088 | 0.0648 | 0.6168 | 0.7800  | 0.8957 | 0.1067 | 0.4282 | 0.6555  | 0.9249 | 0.0544 | 0.6094 | 0.8197  | **0.9210**    | **0.0663**   |
> >
> >
> > While increased inconsistency (lower CS values) may cause SMR to fail in tracking due to unreliable memory, resulting in minor performance degradation, the overall high consistency observed across datasets ensures robust performance in practice. It is precisely this temporal stability that enables SMR to effectively reduce redundant computations while maintaining accuracy. In future work, we will further enhance SMR's mechanism to improve its adaptability under varying conditions.

---

### Official Review · Reviewer_gSJw · 2025-11-01

**Soundness:** 3
**Presentation:** 3
**Contribution:** 3
**Rating:** 6
**Confidence:** 5

**Summary:**

This paper proposes Efficient-SAM2, a post-training acceleration framework to address the significant computational bottleneck of SAM2 in real-time video object segmentation. The authors identify a mismatch between SAM2's dense computation and its inherently sparse perception, highlighting redundancy in the image encoder's background processing and in full-token memory retrieval. To exploit this, the framework introduces two key components: object-aware Sparse Window Routing (SWR) and object-aware Sparse Memory Retrieval (SMR). SWR dynamically routes irrelevant background windows in the encoder to a lightweight shortcut branch, guided by saliency and consistency cues from the previous frame's decoder. SMR leverages temporal consistency by identifying a sparse set of salient memory tokens during their first recollection and reusing this pattern for subsequent frames, drastically reducing memory attention computations.

**Strengths:**

1. The motivation for reducing SAM2's computational overhead is well-grounded and intuitive for video object segmentation
2. The post-training approach is practical, enabling efficient adaptation by leveraging the generalized parameters of the pre-trained SAM2.
3. The method achieves a good speed-performance trade-off, delivering a speedup of nearly 2x while incurring only a minimal and acceptable performance degradation of approximately 1%.

**Weaknesses:**

1. The SWR component is heavily dependent on the previous frame's prediction and salient mask. I think this may cause challenges in some cases, such as rapid motion, abrupt scene cuts, or severe occlusions, where this temporal assumption would be violated.
2. For a video domain paper, the qualitative results with static images are insufficient. Supplemental videos would be significantly stronger to properly demonstrate temporal consistency, failure modes (especially in scenarios mentioned in point 1), and the practical impact of the optimizations.
3. I think an evaluation is needed to determine if the SMR module, which uses a cached saliency pattern, maintains its effectiveness in long video scenarios where significant appearance and context drift are likely.
4. The paper lacks a dedicated discussion of its limitations. This omission leaves the impression that the proposed efficiencies might be confined to easy (or trained) scenarios.

**Questions:**

While the paper presents a valuable approach to making SAM2 more efficient, the design of the SWR and SMR modules appears highly heuristic and is tightly coupled to assumptions about temporal continuity. This raises a significant question: Is SAM2's strong, general-purpose segmentation performance fully preserved? The evaluation is currently confined to standard VOS benchmarks. To truly validate that these heuristics do not compromise the model's robustness, I would ask the authors to provide evaluations on more diverse datasets, particularly on challenging "in-the-wild" videos, which would better test the limits of these heuristic assumptions.

---

> ### Author Response · Authors · 2025-11-19
>
> Thank you for your constructive feedback. We hope our responses provide sufficient clarification.
>
> ---
>
> >W1,W3,Q1: Reliance on the temporal continuity and consistency assumption raises concerns about the performance of SWR and SMR on more challenging tasks.
> >
> Thank you for raising these valuable concerns.
>
> **Concerns about SWR with temporal continuity assumption :**
> While the window router does rely on the previous frame's prediction and saliency, it is important to note that windows assigned as background are not discarded. Instead, they are processed through a shortcut network and aggregated with foreground windows in the final layer, thus allowing the model to maintain a global view throughout the processing pipeline.
>
> Furthermore, SWR specifically designed two strategies to address challenging scenarios such as fast motion and severe distraction: (1) **mask dilation**, which helps capture motion-escaping objects that might move beyond their predicted regions; (2) **involving all three predictions** instead of only the mask with the highest IoU score, which leverages SAM2's inherent ability to perceive and handle distractions. **Ablation studies in Table 4** validate the effectiveness of these strategies.
>
> **Concerns about SMR with temporal consistency assumption :**
> SMR is motivated by the observation of temporal consistency in memory sparse patterns. This observation proves reliable in most cases, and we have conducted statistical analyses across multiple datasets to support this assumption, which serves as an important foundation for ensuring SMR's performance (See table in W2).
>
> **Additional Evaluation on Challenging Benchmarks:**
> In response to your suggestion, we have supplemented our experiments with more challenging benchmarks including MOSEv2[1] and SeCVOS[2], which contain complex real-world scenarios for evaluating in-the-wild tracking capabilities.
>
> - **MOSEv2** features longer video sequences that test the model's long-term tracking capability with specialized evaluation metrics for disappearance-reappearance scenarios: $J\\&\dot{F}_d$ (Disappearance Metric) and $J\\&\dot{F}_r$ (Reappearance Metric).
>
> - **SeCVOS** focuses on diverse complex video scenarios. *Segmented metrics* are designed to examine the model's ability of handle scene changes (SC), including No SC, Single SC, and Multiple SC.
>
> **Results on MOSEv2 Benchmark:**
>
> | Model | Method | IE | MA | $J\\&\dot{F}$ | $J\\&\dot{F}_d$ | $J\\&\dot{F}_r$ | $J\\&F$ |
> |:-------:|:--------:|:----:|:----:|:--------------:|:----------------:|:----------------:|:--------:|
> | SAM2.1-B | Ori | - | - | 43.29 | 60.59 | 20.73 | 44.38 |
> | | ToMe | $\checkmark$ | $\times$ | 20.07 | 48.49 | 2.00 | 20.82 |
> | | **SWR (ours)** | $\checkmark$ | $\times$ | 42.82 | 60.40 | 20.13 | **43.9** |
> | | MemPool | $\times$ | $\checkmark$ | 40.86 | 61.23 | 18.32 | 41.72 |
> | | **SMR (ours)** | $\times$ | $\checkmark$ | 43.44 | 58.04 | 21.94 | **44.56** |
> | | **Efficient-SAM2 (ours)** | $\checkmark$ | $\checkmark$ | 42.23 | 58.10 | 19.54 | **43.36** |
> | SAM2.1-L | Ori | - | - | 49.56 | 63.34 | 27.90 | 50.81 |
> | | ToMe | $\checkmark$ | $\times$ | 20.38 | 61.21 | 1.62 | 20.97 |
> | | **SWR (ours)** | $\checkmark$ | $\times$ | 45.44 | 57.73 | 23.13 | **46.67** |
> | | MemPool | $\times$ | $\checkmark$ | 43.29 | 62.44 | 21.22 | 44.19 |
> | | **SMR (ours)** | $\times$ | $\checkmark$ | 48.70 | 62.19 | 27.27 | **49.98** |
> | | **Efficient-SAM2 (ours)** | $\checkmark$ | $\checkmark$ | 44.96 | 57.05 | 22.04 | **46.20** |
>
>
> **Results on SeCVOS Benchmark:**
>
> | Model | Method | IE | MA | No SC | Single SC | Multi SC | **Overall** |
> |:-------:|:--------:|:----:|:----:|:-------:|:-----------:|:----------:|:-------------:|
> | SAM2.1-B | Ori | - | - | 77.4 | 58.5 | 51.8 | 57.4 |
> | | ToMe | $\checkmark$ | $\times$ | 48.9 | 15.4 | 12.4 | 18.2 |
> | | **SWR (ours)** | $\checkmark$ | $\times$ | 76.9 | 59.3 | 53.4 | **57.5** |
> | | MemPool | $\times$ | $\checkmark$ | 73.3 | 54.8 | 52.3 | 56 |
> | | **SMR (ours)** | $\times$ | $\checkmark$ | 78.4 | 61.1 | 53.3 | **59.0** |
> | | **Efficient-SAM2 (ours)** | $\checkmark$ | $\checkmark$ | 77.7 | 59.3 | 53.7 | **58.3** |
> | SAM2.1-L | Ori | - | - | 79.3 | 57.9 | 52.4 | 58.0 |
> | | ToMe | $\checkmark$ | $\times$ | 49.1 | 17.7 | 15.2 | 20.4 |
> | | **SWR (ours)** | $\checkmark$ | $\times$ | 78.1 | 55.1 | 53.2 | **56.9** |
> | | MemPool | $\times$ | $\checkmark$ | 74.8 | 55.9 | 51.2 | 55.7 |
> | | **SMR (ours)** | $\times$ | $\checkmark$ | 79.5 | 59.1 | 53.0 | **58.7** |
> | | **Efficient-SAM2 (ours)** | $\checkmark$ | $\checkmark$ | 78.4 | 57.8 | 51.3 | **56.9** |
>
>
> [1] MOSEv2: A More Challenging Dataset for Video Object Segmentation in Complex Scenes
>
> [2] SeC: Advancing Complex Video Object Segmentation via Progressive Concept Construction

---

> ### Author Response · Authors · 2025-11-19
>
> >W2: Static qualitative results are insufficient to demonstrate temporal consistency.
> >
> Thanks. To validate the temporal consistency, we quantify cross-temporal saliency patterns using mean cosine similarity (CS) between initial and subsequent attention scores of memory frames. Results show high temporal consistency across all datasets, with mean CS > 0.9 and concentrated distributions (average standard deviation ≤ 0.08 across layers).
>
> | Model | Layer | SA-V test |        |        |         | MOSE   |        |        |         | SeCVOS |        |        |         | DAVIS  |        |        |         | **Avg. mean** | **Avg. std** |
> |:-----:|:-----:|:---------:|:------:|:------:|:-------:|:------:|:------:|:------:|:-------:|:------:|:------:|:------:|:-------:|:------:|:------:|:------:|:-------:|:-------------:|:------------:|
> |       |       | mean      | std    | min    | 5% pct. | mean   | std    | min    | 5% pct. | mean   | std    | min    | 5% pct. | mean   | std    | min    | 5% pct. |               |              |
> | SAM2.1-B+ | 0     | 0.9896    | 0.0089 | 0.8802 | 0.9764  | 0.9818 | 0.0177 | 0.8908 | 0.9418  | 0.9654 | 0.0436 | 0.6580 | 0.8700  | 0.9912 | 0.0151 | 0.7663 | 0.9721  | **0.9820**    | **0.0213**   |
> |       | 1     | 0.9589    | 0.0392 | 0.5836 | 0.8976  | 0.8990 | 0.0772 | 0.6185 | 0.7410  | 0.8791 | 0.1257 | 0.3749 | 0.6052  | 0.9344 | 0.0553 | 0.4045 | 0.8452  | **0.9179**    | **0.0744**   |
> |       | 2     | 0.9990    | 0.0021 | 0.9479 | 0.9970  | 0.9968 | 0.0047 | 0.9597 | 0.9882  | 0.9951 | 0.0079 | 0.9238 | 0.9788  | 0.9988 | 0.0021 | 0.9790 | 0.9959  | **0.9974**    | **0.0042**   |
> |       | 3     | 0.9967    | 0.0103 | 0.7513 | 0.9900  | 0.9882 | 0.0176 | 0.8732 | 0.9554  | 0.9691 | 0.0539 | 0.3047 | 0.8663  | 0.9949 | 0.0137 | 0.7212 | 0.9780  | **0.9872**    | **0.0239**   |
> | SAM2.1-L | 0     | 0.9998    | 0.0005 | 0.9882 | 0.9992  | 0.9994 | 0.0010 | 0.9950 | 0.9973  | 0.9980 | 0.0044 | 0.9074 | 0.9901  | 0.9997 | 0.0012 | 0.9763 | 0.9987  | **0.9992**    | **0.0018**   |
> |       | 1     | 0.9848    | 0.0195 | 0.7262 | 0.9554  | 0.9588 | 0.0378 | 0.8010 | 0.8789  | 0.9437 | 0.0680 | 0.5650 | 0.7911  | 0.9605 | 0.0360 | 0.6057 | 0.8926  | **0.9620**    | **0.0403**   |
> |       | 2     | 0.9639    | 0.0333 | 0.6748 | 0.9052  | 0.9301 | 0.0577 | 0.6309 | 0.8148  | 0.9005 | 0.1052 | 0.4214 | 0.6569  | 0.9342 | 0.0550 | 0.5086 | 0.8315  | **0.9322**    | **0.0628**   |
> |       | 3     | 0.9544    | 0.0394 | 0.4821 | 0.8828  | 0.9088 | 0.0648 | 0.6168 | 0.7800  | 0.8957 | 0.1067 | 0.4282 | 0.6555  | 0.9249 | 0.0544 | 0.6094 | 0.8197  | **0.9210**    | **0.0663**   |
>
> This temporal stability enables SMR to reduce redundant computations while maintaining accuracy. Although lower CS may cause slight performance degradation due to unreliable memory cues, the consistently high CS across datasets ensures robust practical performance. We will optimize SMR's mechanisms for enhanced adaptability in future work.
>
> Video demonstrations are essential for a video domain paper. We will provide qualitative results in video format when the code is publicly released.
>
>
> >W4: Concerns about efficiency limitation.
> >
> Thank you for this valuable feedback. **In Section 4.3, we present the reduction ratio (sparsity) of windows and memory tokens** to evaluate the efficiency optimization of the proposed method across different datasets. Here, we provide a more detailed clarification of the efficiency benefits:
>
> - For SMR: The sparsity ratio can be pre-configured, ensuring controllable efficiency benefits. Results across various complex datasets show that SMR effectively maintains performance even at high memory sparsity (0.678).
>
> - For SWR: **Its adaptive selection strategy adjusts window sparsity based on task complexity and object scale.** Notably, challenging tracking tasks typically feature smaller target scales and stronger environmental interference, which naturally ensures a certain degree of window sparsity. Table 5 reports mean window sparsity (mWS) across datasets, confirming broad efficiency gains. To further evaluate the performance-efficiency trade-off, we tested two additional challenging datasets:
>
> | Model | mWS on MOSEv2 | mWS on SeCVOS |
> |:-----:|:------:|:------:|
> | SAM2.1-B+ | 0.79 | 0.63 |
> | SAM2.1-L | 0.76 | 0.62 |
>
> While SWR tends to allocate slightly more windows as foreground in more complex scenarios, this behavior is consistent with our design rationale of using cumulative attention rather than a fixed number for window allocation, enabling the adaptive advantage of window routing.

---

### Official Review · Reviewer_DdRT · 2025-11-02

**Soundness:** 3
**Presentation:** 3
**Contribution:** 3
**Rating:** 6
**Confidence:** 5

**Summary:**

The paper aims to reduce the high inference cost of SAM2 in video segmentation. Specifically, they propose two modules: Object-aware Sparse Window Routing (SWR), which skips background windows in the image encoder based on object masks and saliency, and Object-aware Sparse Memory Retrieval (SMR), which selects only salient memory tokens and reuses their mask across frames. Together, these modules accelerate SAM2 by up to 1.75× with minimal accuracy loss on benchmarks such as SA-V, DAVIS, and MOSE.

**Strengths:**

1. The paper makes a solid technical contribution by streamlining the SAM2 model. The object-aware pruning of the image encoder and the introduction of a background shortcut for non-foreground patches are both clever ideas that substantially reduce computation.

2. The ablation study is comprehensive. It not only analyzes the proposed components in isolation but also integrates other efficient methods (e.g., ToME) into their framework for comparison, which provides valuable insights.

**Weaknesses:**

1. The proposed routing mechanism heavily relies on the assumption of temporal consistency in video streams, meaning that no significant camera shaking or viewpoint shift occurs. This limits the method’s applicability in real-world scenarios with dynamic motion. It would be interesting to see comparisons with SAM2 on datasets such as MOSEv2[1] and SeCVOS[2], which feature frequent viewpoint transitions.




2. Performing grid search on only one benchmark is not sufficient to demonstrate robustness. It would strengthen the paper to include additional grid search curves across multiple benchmarks (in Figure 5).







[1] MOSEv2: A More Challenging Dataset for Video Object Segmentation in Complex Scenes
[2] SeC: Advancing Complex Video Object Segmentation via Progressive Concept Construction

**Questions:**

1. What is the specific layer index after which the router divides foreground and background tokens? How many subsequent modules benefit from reduced computation? It would be helpful to include an ablation varying the routing layer index to assess its impact.

2. How is speedup defined in this paper? It would be clearer to disclose it in two aspects—FLOPs reduction and throughput improvement—to give a more complete view of efficiency.

3. What is the intuition behind using two different temporal intervals (∆t =1,5) in the experiments? It seems the performance difference might largely stem from the frame rate (FPS) of the original benchmark. For high-FPS datasets like SA-V, increasing the interval could naturally yield greater gains since the memory bank contains more diverse frames.

4. For SMR, have you evaluated a variant that selects memory frames only when object presence is confident (akin to SAM2Long’s strategy)? It would be informative to report the gain from this filtering.

---

> ### Author Response · Authors · 2025-11-19
>
> Thanks for your thoughtful comments and valuable suggestions. We hope our responses below adequately clarify the issues raised.
>
> ---
>
> >W1: Reliance on the temporal consistency assumption raises concerns about performance on more challenging tasks.
> >
> Thanks. Although the window router relies on the previous frame’s prediction and saliency, windows assigned as background in the current frame are not discarded but rather processed by a shortcut network, thus maintaining a global view.
>
> Moreover, the window routing mechanism takes into account scenarios involving fast motion and severe occlusions, and proposes two strategies: (1) mask dilation, which helps capture motion-escaping objects; (2) involving all three predictions, which leverages SAM2's inherent ability to perceive and handle distractions. **Ablation studies in Table 4 show the effectiveness of these strategies.**
>
> To evaluate our methods on videos with frequent viewpoint transitions, we follow your recommendation and include results on MOSEv2 and SeCVOS.
>
> **Results on MOSEv2 Benchmark:**
>
> | Model | Method | IE | MA | $J\\&\dot{F}$ | $J\\&\dot{F}_d$ | $J\\&\dot{F}_r$ | $J\\&F$ |
> |:----:|:----:|:----:|:----:|:-----:|:----:|:----:|:----:|
> | SAM2.1-B | Ori | - | - | 43.29 | 60.59 | 20.73 | 44.38 |
> | | ToMe | $\checkmark$ | $\times$ | 20.07 | 48.49 | 2.00 | 20.82 |
> | | **SWR (ours)** | $\checkmark$ | $\times$ | 42.82 | 60.40 | 20.13 | **43.9** |
> | | MemPool | $\times$ | $\checkmark$ | 40.86 | 61.23 | 18.32 | 41.72 |
> | | **SMR (ours)** | $\times$ | $\checkmark$ | 43.44 | 58.04 | 21.94 | **44.56** |
> | | **Efficient-SAM2 (ours)** | $\checkmark$ | $\checkmark$ | 42.23 | 58.10 | 19.54 | **43.36** |
> | SAM2.1-L | Ori | - | - | 49.56 | 63.34 | 27.90 | 50.81 |
> | | ToMe | $\checkmark$ | $\times$ | 20.38 | 61.21 | 1.62 | 20.97 |
> | | **SWR (ours)** | $\checkmark$ | $\times$ | 45.44 | 57.73 | 23.13 | **46.67** |
> | | MemPool | $\times$ | $\checkmark$ | 43.29 | 62.44 | 21.22 | 44.19 |
> | | **SMR (ours)** | $\times$ | $\checkmark$ | 48.70 | 62.19 | 27.27 | **49.98** |
> | | **Efficient-SAM2 (ours)** | $\checkmark$ | $\checkmark$ | 44.96 | 57.05 | 22.04 | **46.20** |
>
>
> **Results on SeCVOS Benchmark:**
>
> | Model | Method | IE | MA | No SC | Single SC | Multi SC | **Overall** |
> |:-----:|:----:|:----:|:----:|:----:|:----:|:----:|:----:|
> | SAM2.1-B | Ori | - | - | 77.4 | 58.5 | 51.8 | 57.4 |
> | | ToMe | $\checkmark$ | $\times$ | 48.9 | 15.4 | 12.4 | 18.2 |
> | | **SWR (ours)** | $\checkmark$ | $\times$ | 76.9 | 59.3 | 53.4 | **57.5** |
> | | MemPool | $\times$ | $\checkmark$ | 73.3 | 54.8 | 52.3 | 56 |
> | | **SMR (ours)** | $\times$ | $\checkmark$ | 78.4 | 61.1 | 53.3 | **59.0** |
> | | **Efficient-SAM2 (ours)** | $\checkmark$ | $\checkmark$ | 77.7 | 59.3 | 53.7 | **58.3** |
> | SAM2.1-L | Ori | - | - | 79.3 | 57.9 | 52.4 | 58.0 |
> | | ToMe | $\checkmark$ | $\times$ | 49.1 | 17.7 | 15.2 | 20.4 |
> | | **SWR (ours)** | $\checkmark$ | $\times$ | 78.1 | 55.1 | 53.2 | **56.9** |
> | | MemPool | $\times$ | $\checkmark$ | 74.8 | 55.9 | 51.2 | 55.7 |
> | | **SMR (ours)** | $\times$ | $\checkmark$ | 79.5 | 59.1 | 53.0 | **58.7** |
> | | **Efficient-SAM2 (ours)** | $\checkmark$ | $\checkmark$ | 78.4 | 57.8 | 51.3 | **56.9** |
>
> > W2: Grid search for hyperparameter on more benchmark
> >
> Thanks. We conduct grid search on additional datasets, showing the robustness of our method to hyperparameter variations.
>
> | $\tau$ | 0.1 | 0.2 | 0.3 | 0.4 | 0.5 | 0.6 | **0.7** | 0.8 | 0.9 | baseline |
> |:-:|:---:|:---:|:---:|:---:|:---:|:---:|:---:|:---:|:---:|:--------:|
> | **$J\\&F$ on MOSE** | 71.1 | 71.1 | 71.1 | 71.3 | 70.9 | 71.2 | 71.2 | 71.5 | 72.3 | 73.6 |
> | Window Sparsity | 0.77 | 0.76 | 0.76 | 0.75 | 0.74 | 0.73 | 0.72 | 0.68 | 0.61 | 0.00 |
> | **$J\\&F$ on SeCVOS** | 55.2 | 57.1 | 56.8 | 56.8 | 57.0 | 57.9 | 57.5 | 56.9 | 58.4 | 57.4 |
> | Window Sparsity | 0.76 | 0.74 | 0.72 | 0.71 | 0.69 | 0.66 | 0.63 | 0.58 | 0.48 | 0.00 |
>
>
> | $s$ | 0.1 | 0.3 | 0.5 | 0.7 | 0.8 | 0.85 | 0.9 | **0.95** | 0.98 | Baseline |
> |:-:|:----:|:----:|:----:|:----:|:----:|:----:|:----:|:----:|:-----:|:--------:|
> | **$J\\&F$ on MOSE** | 73.65 | 73.68 | 73.65 | 73.5 | 73.48 | 73.45 | 73.4 | 73.2 | 72.99 | 73.6 |
> | Token Sparsity | 0.07 | 0.21 | 0.36 | 0.50 | 0.57 | 0.61 | 0.64 | 0.68 | 0.70 | 0 |
> | **$J\\&F$ on SeCVOS** | 57.6 | 57.5 | 57.8 | 57.7 | 57.4 | 58.0 | 58.1 | 59.0 | 53.7 | 57.4 |
> | Token Sparsity | 0.07 | 0.21 | 0.36 | 0.50 | 0.57 | 0.61 | 0.64 | 0.68 | 0.70 | 0 |
>
> | $\theta_{obj}$          | 1      | 3      | 5      | 7      | 9      |
> |:---:|:----:|:----:|:----:|:---:|:---:|
> | **$J\\&F$ on SAV_test**     | 74.7   | 75.1   | 75     | 75.2   | 75     |
> | Window Sparsity         | 0.781  | 0.781  | 0.781  | 0.781  | 0.781  |
> | **$J\\&F$ on SeCVOS**       | 57.8   | 57.6   | 57.5   | 58     | 58.1   |
> | Window Sparsity         | 0.632  | 0.632  | 0.630  | 0.631  | 0.629  |
> | **$J\\&\dot{F}$ on MOSEv2**      | 42.11  | 42.26  | 42.2   | 42.82  | 42.4   |
> | Window Sparsity         | 0.796  | 0.795  | 0.795  | 0.797  | 0.797  |

---

> ### Author Response · Authors · 2025-11-19
>
> > Q1: Router layer settings and ablation study.
> >
> Thank you for the suggestion. SAM2’s backbone adopts a multi-stage design with three stages (Stage0, Stage1, Stage2). In our default configuration, the window router is placed after Stage1, so only Stage2, where most computation occurs, benefits from computation reduction. Specifically:
>
> - Base+: router before Layer 6; Layers 6–19 benefit; Layer 20 as aggregation.
> - Large: router before Layer 9; Layers 9–42 benefit; Layer 43 as aggregation.
>
> **We conducted an ablation on the router’s placement in Appendix E, and add results with more configuration here.** Placing the router in early stages (Stage0 or Stage1) is not cost-effective due to their limited layers and direct connection to the mask decoder. Inserting the router in the middle of Stage2 maintains accuracy but offers less efficiency improvement. For further details and comprehensive analysis, see Appendix E.
>
> | Model | Layers before Router | Routing layers | Aggregation layer| $J\\&F$ (Identity Shortcut) | $J\\&F$ (Trained Shortcut) |
> |:---:|:---:|:---:|:---:|:---:|:---:|
> | SAM2.1-B+(77.7) | 0 | 1-19 | 20 | 69.3 | 74.0 |
> |  | **0-5** | **6-19** | **20** | **73.0** | **75.0** |
> |  | 0-12 | 13-19 | 20 | 73.1 | 76.1 |
> | SAM2.1-L(79.8) | 0 | 1-42 | 43 | 77.2 | 77.4 |
> |  | **0-5** | **9-42** | **43** | **75.7** | **79.0** |
> |  | 0-23 | 24-42 | 43 | 79.9 | 80.0 |
>
> >Q2: Clarification and supplement about speedup.
> >
> Thanks for your valuable suggestion. The reported speedup refers to the improvement in measured throughput, i.e., the ratio of FPS (frames per second). **We provide a detailed description of the speed benchmarking protocol in Appendix B.**
>
> FLOPs is indeed an important efficiency metric worth considering. We supplement the analysis with GFLOPs reduction for Efficient-SAM2 (with window sparsity of 0.75 and memory sparsity factor $s = 0.95$):
>
> | Models | Image encoder GFLOPs | Memory attention GFLOPs |
> |:---:|:---:|:---:|
> | SAM2.1-B+ | 659.6 | 300.1 |
> | **Efficient-SAM2.1-B+** | 296.2 ($\downarrow$ 2.23 $\times$) | 122.0($\downarrow$ 2.46 $\times$) |
> | SAM2.1-L | 1798 | 300.1 |
> | **Efficient-SAM2.1-L** | 541.9 ($\downarrow$ 3.32 $\times$) | 122.0($\downarrow$ 2.46 $\times$) |
>
>
> >Q3: Intuition of using different memory intervals.
> >
> Thank you for this nice question. Using different memory intervals (∆t=1 and ∆t=5) allows for a more thorough evaluation of SMR. As you pointed out, increasing the memory interval improves baseline performance on dynamic datasets like SA-V by enriching memory diversity, indicating that ∆t=1 underutilizes the memory mechanism's potential. Moreover, Larger intervals allow a memory frame persists longer in the queue and participates in more computation steps, which poses a challenge to temporal consistency of memory sparse pattern.
>
> Surprisingly, results show that SMR remains robust with larger memory intervals. This confirms that the "sparse pattern consistency" persists over extended time spans, allowing SMR to maintain effectiveness even as memory become more informative.
>
> >Q4: Evaluate method with selected memories.
> >
> Thanks for your valuable suggestion. Selecting memory frames to ensure a more reliable and informative memory bank is indeed a valuable experiment that enables a more effective evaluation of SMR's performance.
>
> Following your recommendation, we have implemented memory frame filtering according to SAM2Long's strategy (denoted as MF). Specifically, we retain only frames where the object is present with an IoU score greater than 0.3. The results show that MF improves SAM2 performance, and SMR effectively preserves these gains. This indicates that SMR's sparse strategy is equally effective for selected memory, which further validates its robustness.
>
> | Method | $J\\&F$ on SA-V test | $J\\&\dot{F}$ on MOSEv2 |
> |:------:|:----------------:|:--------------:|
> | Ori | 77.7 | 43.29 |
> | MF | 80 | 47.99 |
> | MF+MemPool | 75.4 | 43.98 |
> | **MF+SMR** | **79** | **47.58** |

---

### Author Response · Authors · 2025-12-03
**Rebuttal Summary of Submission 4947**

We sincerely thank all reviewers for their constructive suggestions and valuable insights, below is a summary of our response.

We are encouraged by the reviewers' recognition of our innovation and technical contribution to SAM2's efficiency:
1. **Novelty and Methodology.** The motivation and innovation for streamlining SAM2 are well-grounded and intuitive (gSJw, FsUz), with the routing and shortcut ideas notably clever (DdRT). The modular post-training scheme is both practical and scalable, incurring negligible overhead (gSJw, FsUz).
2. **Comprehensive Validation.** The component-wise comparison offers valuable insights into effectiveness (DdRT), highlighting an excellent speed–performance trade-off (gSJw) and strong scalability to larger models (FsUz).

We have carefully considered all reviewers' suggestions, performed additional experiments, and provided detailed clarifications to address the raised concerns. A summary follows.

1. **Requests to evaluate the method's competence in more complex scenarios.**
We added evaluations on two more challenging and comprehensive benchmarks (MOSEv2 and SeCVOS), showing strong adaptability to diverse and difficult cases. (DdRT W1, gSJw W1 & Q1, FsUz Q2).

2. **Concerns about claimed temporal continuity and consistency.**
   - **Temporal continuity assumption:** While this assumption is largely reliable and well-aligned with SAM2's nature, our method employs dedicated strategies (shortcut network, mask dilation, and include alternative predictions) to handle edge cases. Results on hard benchmarks (DdRT Q1, gSJw W1 & Q1, FsUz Q2) and ablation studies (Table 3, 4) demonstrate the effectiveness of these designs.
   - **Temporal consistency of memory saliency pattern:** (1) We performed statistical analysis to empirically validate the generality of this observation (FsUz Q6); (2) We evaluated our method on memory-selected variants of SAM2, further validating its scalability (DdRT Q4).

3. **Requests for details about efficiency improvement.**
    - **Efficiency gain:** We provided a detailed ablation study on SWR speedup layers (Appendix E), and reported additional settings for further refinement (DdRT Q1). We present sparsity analysis to evaluate cross-datasets efficiency gain (Section 4.3), and provided additional results on more complex datasets (gSJw W4). These results enhance the validation of method's efficiency benefits.
    - **Speed benchmark:** We presented detailed description of the speed benchmarking protocol in Appendix B, and added FLOPs reduction results in the rebuttal to give a more complete view of efficiency (DdRT Q2).

Although we did not have enough time to discuss fully with the reviewers, we believe our responses substantially address the raised concerns, and the additional experiments have strengthened our work considerably. We sincerely thank the reviewers for their insightful feedback and the ACs for their dedication and guidance throughout the review process.

---

### Meta-Review · Area_Chair_mShb · 2025-12-29

**Summary:**

The paper proposes Efficient-SAM2, a post-training acceleration framework for SAM2 with two core contributions: Sparse Window Routing (SWR) for the image encoder and Sparse Memory Retrieval (SMR) for memory attention. The reviewers collectively acknowledge the practical value of achieving speedup with minimal accuracy loss and appreciate the comprehensive ablation studies.

The main concerns of reviewers are:

1. The method's reliance on previous-frame predictions generalizes to challenging real-world scenarios (e.g., fast motion, scene cuts, occlusions).

2. Initial experiments were confined to standard VOS benchmarks, lacking evidence on in-the-wild challenging videos.

3. Static image results are inadequate for a video domain paper; temporal behavior and failure modes need demonstration.

4. Reviewer FsUz reported code execution failures.

**Reviewer Concerns:**

During the rebuttal, authors add MOSEv2 and SeCVOS results, showing competitive performance on challenging scenarios. As for hyperparameter sensitivity, the authors perform a grid search extended to MOSE, SeCVOS, MOSEv2 with multiple parameters. The authors also provide comprehensive CS statistics (mean > 0.9) across 4 datasets and multiple layers. The authors also evaluate SAM2Long-style filtering, showing SMR preserves gains. As for the code reproducibility, the authors claim successful execution on 3 machines and provide environment specs, but cannot verify without reviewer confirmation.

**Reviewer Scores:**

Most technical questions of reviewers DdRT, gSJw, and FsUz are thoroughly answered. The authors delivered a strong rebuttal with substantial new experiments (2 additional challenging benchmarks, extensive grid searches, statistical validation of temporal consistency, GFLOPs analysis). The core technical contribution is sound, and the efficiency gains are meaningful for practical deployment.

---

### Decision · Program_Chairs · 2026-01-26

Accept (Poster)